# Identification of phosphatases that dephosphorylate the co-chaperone BAG3

Thomas Kokot[1,2] , Johannes P Zimmermann[3] , Yamini Chand[1,2] , Fabrice Krier[1,2] , Lena Reimann[4] , Laura Scheinost[1,2] , Nico Höfflin[1,2], Alessandra Esch[5], Jörg Höhfeld[5], Bettina Warscheid[3,4] , Maja Köhn[1,2,5]

**The co-chaperone BAG3 plays critical roles in maintaining cellular proteostasis. It associates with 14-3-3 proteins during the trafficking of aggregation-prone proteins and facilitates their degradation through chaperone-assisted selective autophagy in cooperation with small heat shock proteins. Although reversible phosphorylation regulates BAG3 function, the involved phosphatases remain unknown. Here, we used affinity purification mass spectrometry to identify phosphatases that target BAG3. Of the hits, we evaluated the involvement of protein phosphatase-1 (PP1) using chemical inhibitors and activators in in vitro and cellular approaches. Our results demonstrate that PP1 can dephosphorylate BAG3-pS136 in cells and counteract 14-3-3γ association with BAG3 at this motif. Furthermore, protein phosphatase-5 (PP5) co-enriched with proteostasis-related proteins, and it has the capacity to dephosphorylate a BAG3 phosphorylation-site cluster regulating the interaction of BAG3 with small heat shock proteins and BAG3-mediated protein degradation. Our findings provide new insights into the regulation of BAG3 by phosphatases. This paves the way for future research focused on the precise control of BAG3 function through its regulatory proteins, potentially holding therapeutic promise for diseases characterized by disrupted proteostasis.**

## Introduction

The functional proteome constitutes the cornerstone of cellular activity, with protein interactions orchestrating nearly all cellular processes. Disruptions in protein function often lead to pathway perturbations, driving cellular impairment and the onset of various diseases (Hipp et al, 2019; Bludau & Aebersold, 2020). To maintain proteome integrity, cells have evolved a multifaceted proteostasis system involving protein degradation, refolding, and synthesis (Chen et al, 2011; Toyama & Hetzer, 2013; Ciechanover & Kwon, 2017;

Dikic & Elazar, 2018; Hipp et al, 2019). Molecular chaperones and their co-chaperones play a central role in maintaining functional proteostasis (Balchin et al, 2016; Höhfeld et al, 2021). In this context, much attention has been directed towards the co-chaperone Bcl2-associated athanogene 3 (BAG3) because of its involvement in multiple proteostasis pathways and stress responses (Pagliuca et al, 2003; Chen et al, 2004; Rosati et al, 2007; Ammirante et al, 2010; Basile et al, 2011; Ulbricht et al, 2015; Höhfeld et al, 2021). Impairment of BAG3's specific function to clear disease-related aggregate-prone proteins through chaperone-assisted selective autophagy (CASA) was implicated in various diseases, including age-related disorders such as Alzheimer's disease (Lei et al, 2015), Huntington's disease (Carra et al, 2008; Fuchs et al, 2009), cancer (De Marco et al, 2018; Kirk et al, 2021), and (cardio-)myopathies (Kimura et al, 2021; Qu et al, 2022). BAG3 engages a diverse array of cellular pathways by interacting with a broad spectrum of proteins. Interaction with 14-3-3 proteins allows BAG3 to facilitate the concentration of aggregation-prone proteins in aggresomes (Xu et al, 2013) and cooperation with Hsp70 family members and small heat shock proteins such as HspB8 governs the selection of protein clients for degradation through CASA (Arndt et al, 2010). Despite establishing the roles of BAG3 in numerous cellular processes, our understanding of its spatiotemporal coordination within these pathways, and the detailed interaction with proteins remains limited. It was reported that the correct spatiotemporal function of BAG3 can depend on its phosphorylated residues (Li et al, 2013a; Xu et al, 2013; Kim et al, 2016; Luthold et al, 2021). BAG3 is a partially intrinsically disordered protein with multiple phosphorylation sites (p-sites, Fig 1A), of which only a few have been functionally characterized. For instance, BAG3-phosphoserine(pS)136, as part of a 14-3-3 binding motif, is known to mediate the targeting of aggregated proteins to the aggresome (Xu et al, 2013) and has been observed to be phosphorylated during infection (Bouhaddou et al, 2020) and in cancer (Zhou et al, 2013; Mertins et al, 2016). Another set of phosphorylation sites, including pS284, phosphothreonine(pT)285, pS289, and pS291, are not only dephosphorylated under mechanical stress (Hoffman et al, 2015; Potts et al, 2017; Ottensmeyer et al, 2024)

[1]Institute of Biology III, Faculty of Biology, University of Freiburg, Freiburg im Breisgau, Germany   [2]Signalling Research Centres BIOSS and CIBSS, University of Freiburg, Freiburg im Breisgau, Germany   [3]Biochemistry II, Theodor-Boveri-Institute, University of Würzburg, Würzburg, Germany   [4]Biochemistry and Functional Proteomics, Institute of Biology II, Faculty of Biology, University of Freiburg, Freiburg, Germany   [5]Institute for Cell Biology, University of Bonn, Bonn, Germany

Correspondence: mkoehn@uni-bonn.de

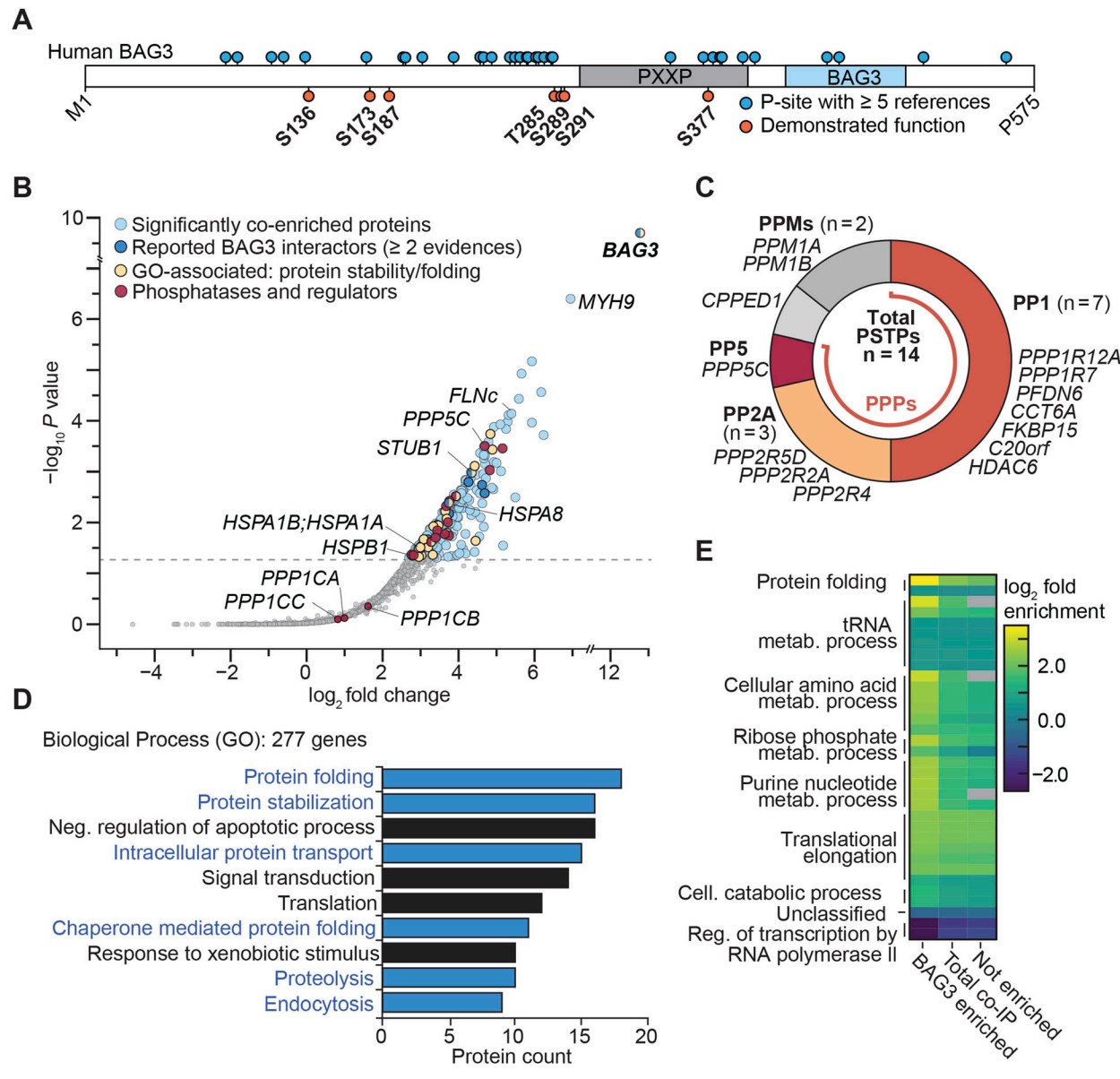

**Figure 1.  BAG3 is a part of the protein maintenance network involving phosphatases.**
**(A)** Linear illustration of human BAG3 protein sequence with reported p-sites annotated. **(B)** Co-IP of FLAG-BAG3 from HEK293 cells revealed significantly enriched proteins, assessed using the rankprod method (n = 4), with a minimum fivefold increase and a $P$-value ≤ 0.05. Enriched proteins in the sample are marked in respective colors for subcategories. Selected hits are annotated with their gene name. A false discovery rate of 5% is indicated by a dashed line. **(C)** Enriched phosphatases and respective phosphatase regulators from the BAG3 co-IP sample were assigned to their respective (super-)families of human phosphatases. The phosphoprotein phosphatase family is colored; other Ser/Thr-specific phosphatases are in shades of gray. Gene names are listed next to the corresponding assigned phosphatase identifications. PPMs, metal-dependent protein phosphatases; PSTPs, protein serine/threonine-specific phosphatases. **(A, D)** Gene ontology enrichment analysis of significantly enriched genes from the BAG3 co-IP sample (n = 277) in (A). The 10 most abundant biological process GO terms are displayed as bars, ranked based on protein counts. Biological processes related to BAG3's role in protein homeostasis are presented in blue. **(E)** Gene ontology enrichment analysis of the co-IP sample compared with overall human gene expression, displayed as a heatmap. Fold enrichment was calculated for the BAG3 enriched, not enriched, and total co-IP sets, with overall GO categories annotated at the respective cluster. Statistical significance was determined using Fisher's test ($P$-value < 0.05) with the Bonferroni correction implemented in the PANTHER database. Non-significant changes are colored gray within the heatmap.

but are also associated with cell shape remodeling (Luthold et al, 2021) and linked to BAG3 homeostasis and cancer (Kim et al, 2016; Zhou et al, 2020).

Although reversible phosphorylation commonly serves as a dynamic mechanism for regulating protein function, pinpointing the specific kinase and phosphatase of a p-site can be challenging.

Currently, only kinases for a few sites have been identified for BAG3, with no phosphatase assigned to its p-sites. In humans, the major family of phosphatases responsible for pS- and pT-site dephosphorylation are the phosphoprotein phosphatases (PPPs) (Li et al, 2013b). Most of these PPPs assemble into multimeric holoenzymes, comprising a catalytic subunit along with regulatory subunits,

allowing them to precisely carry out their function (Kokot & Köhn, 2022). However, because of this intricate regulation, assigning specific phosphatases to p-sites is often challenging and labor-intensive (Fahs et al, 2016; Needham et al, 2019). Nonetheless, determining the regulators of BAG3 is essential for gaining a deeper understanding of its mechanisms.

In this study, we used a combination of mass spectrometric (MS) techniques, small modulator screens, and biochemical enzymatic as well as cellular assays to pinpoint PPPs capable of dephosphorylating BAG3. We identified protein phosphatase-1 (PP1) as a phosphatase of BAG3-pS136 that can regulate the interaction of 14-3-3γ with BAG3. In addition, we found that protein phosphatase-5 (PP5) binds to and can dephosphorylate the cluster of BAG3 p-sites comprising pS284, pT285, pS289, and pS291, thereby regulating BAG3-chaperone complex formation and CASA activity. Taken together, our results highlight the role of PPPs in mediating specific BAG3 protein–protein interactions (PPIs) and processes, contributing to a deeper understanding of the complex protein network governing BAG3 functions within the cell.

# Results

## The BAG3 interactome contains several phosphatases and their regulatory subunits

To identify potential phosphatases involved in the regulation of BAG3, we transfected HEK293 cells with either triple-FLAG–tagged human BAG3 or the triple-FLAG tag that served as the control (Fig S1A). After cell lysis, we carried out single-step affinity enrichment followed by mass spectrometry (AP-MS) using anti-FLAG beads to delineate the BAG3 interactome from HEK293 cells. High-confidence interactors of BAG3 were defined by using an enrichment factor of more than fivefold and using a P-value of 0.05, using the rank sum method (Breitling & Herzyk, 2005). This approach allowed us to narrow down the list of detected proteins to a high-quality set of putative 263 interactors (Fig 1B, Table S1). As a ratification of a robust interactome, we found that 14 of the enriched proteins are listed in the BioGrid database of reported BAG3 interactors with more than n ≥ 2 evidences (Fig 1B) (Stark et al, 2006). These included the WW domain-binding protein 2 (WBP2, Taipale et al, 2014) and components associated with the BAG3-mediated CASA pathway, such as HSPA8 (Takayama et al, 1999) and STUB1 (Arndt et al, 2010; Zhang et al, 2022). Furthermore, we found filamin-C (FLNc), a CASA-client protein (Arndt et al, 2010; Reimann et al, 2020) to be enriched, and observed the presence of (co-)chaperones and heat shock proteins (HSPs; HSPA1A/B, HSPB1, CCTs), which were reported to be involved in BAG3's function in protein maintenance (Fig 1B) (Chen et al, 2013; Taipale et al, 2014; Meister-Broekema et al, 2018; Zhang et al, 2022). We identified phosphatases and their associated regulatory subunits, which were classified into their respective (sub-)families, as illustrated in Fig 1C. Notably, most of the identified phosphatases and regulatory subunits belong to the PPP family (12 of 14). Specifically, our analysis revealed the enrichment of seven PP1-regulatory subunits in association with BAG3, whereas no catalytic subunit isoforms (PPP1CA/B/C) were

observed to be significantly enriched (see Fig 1B). Similarly, three regulatory subunits of PP2A were detected in association with BAG3; yet again, no catalytic subunit isoforms (PPP2CA/PPP2CB) were identified. Notwithstanding this, the PP5 catalytic subunit (PPP5C) emerged as one of the most enriched phosphatases in the dataset. PP5 is one of the few PPP family members that exist as a multidomain protein composed of the catalytic domain (PP5c) and its autoinhibitory, chaperone-binding tetratricopeptide repeat (TPR) domain, and not as a holoenzyme. Pathway enrichment analysis of gene ontology (GO) terms of the co-enriched BAG3 sample demonstrated that the most abundant terms were biological processes related to proteostasis, such as protein folding and protein stabilization (Fig 1D), as well as intracellular protein transport, proteolysis, and endocytosis, which can be connected to BAG3's involvement in macroautophagy (Höhfeld et al, 2021). Comparison of the enriched genes to overall expression of human genes available in the PANTHER database (PANTHER17.0, reference dataset based on UniProt Release 2022_02) (Mi et al, 2013) revealed that protein folding (11-fold) and cellular amino acid metabolic processes (fivefold) were found to be most enriched (Fig 1E).

## BAG3-S136 phosphorylation is affected by CalyculinA treatment and is a substrate of PP1 catalytic subunit in vitro

Several phosphatases and regulatory subunits were identified in the BAG3 interactome. Because BAG3 is a multi-phosphorylated protein, we hypothesized that phosphatases dephosphorylate BAG3 at specific sites to rapidly modulate its functions. To this end, we focused on BAG3 p-sites with described functions (Fig 1A). To explore p-site sequence motifs and their conservation, we conducted a multiple sequence alignment using BAG3 orthologs from 377 vertebrate species (Fig S1B). The MUSCLE alignment showed a high identity score of 0.96 for the SP motif (single-letter amino acid code) of the functional BAG3-S136 site within the RSQS(P) 14-3-3 protein binding motif (Fig S1B). Thus, we decided to pinpoint a phosphatase targeting this specific BAG3 p-site because of (i) its conservation across vertebrates (Fig S1C) and (ii) its proposed role in modulating interactions with 14-3-3γ (Xu et al, 2013). To investigate the regulation of BAG3-pS136 in cells, we used A7r5 rat smooth muscle cells, as they exhibit higher endogenous BAG3 expression levels compared with HEK293 cells used for the MS-approach (Fig S2A). Moreover, the adjacent sequence surrounding the p-site is identical between rat and human (Fig S2B). To assess BAG3-pS136 regulation by members of the PPP family, we tested its responsiveness to generic PPP inhibitors. For this, we monitored the effects of Calyculin A (CalA) and Okadaic acid (OA), both of which exhibit nanomolar potency in vitro. Notably, whereas CalA broadly targets PPPs, including PP1 and PP2A, OA is a more potent inhibitor of PP2A-like phosphatases (PP2A, PP4, PP6) (Zhang et al, 2021). The overall efficacy of the phosphatase inhibitors was verified using a generic pS/pT antibody (Fig 2A). Immunoblot analysis of BAG3-pS136 phosphorylation after CalA and OA treatment revealed sensitivity of BAG3-pS136 to CalA but not OA (Fig 2A). This demonstrates the involvement of PPPs in p-site regulation and suggests the regulation through PP1 over PP2A (PP4, PP6) in this setting (Swingle et al, 2007).

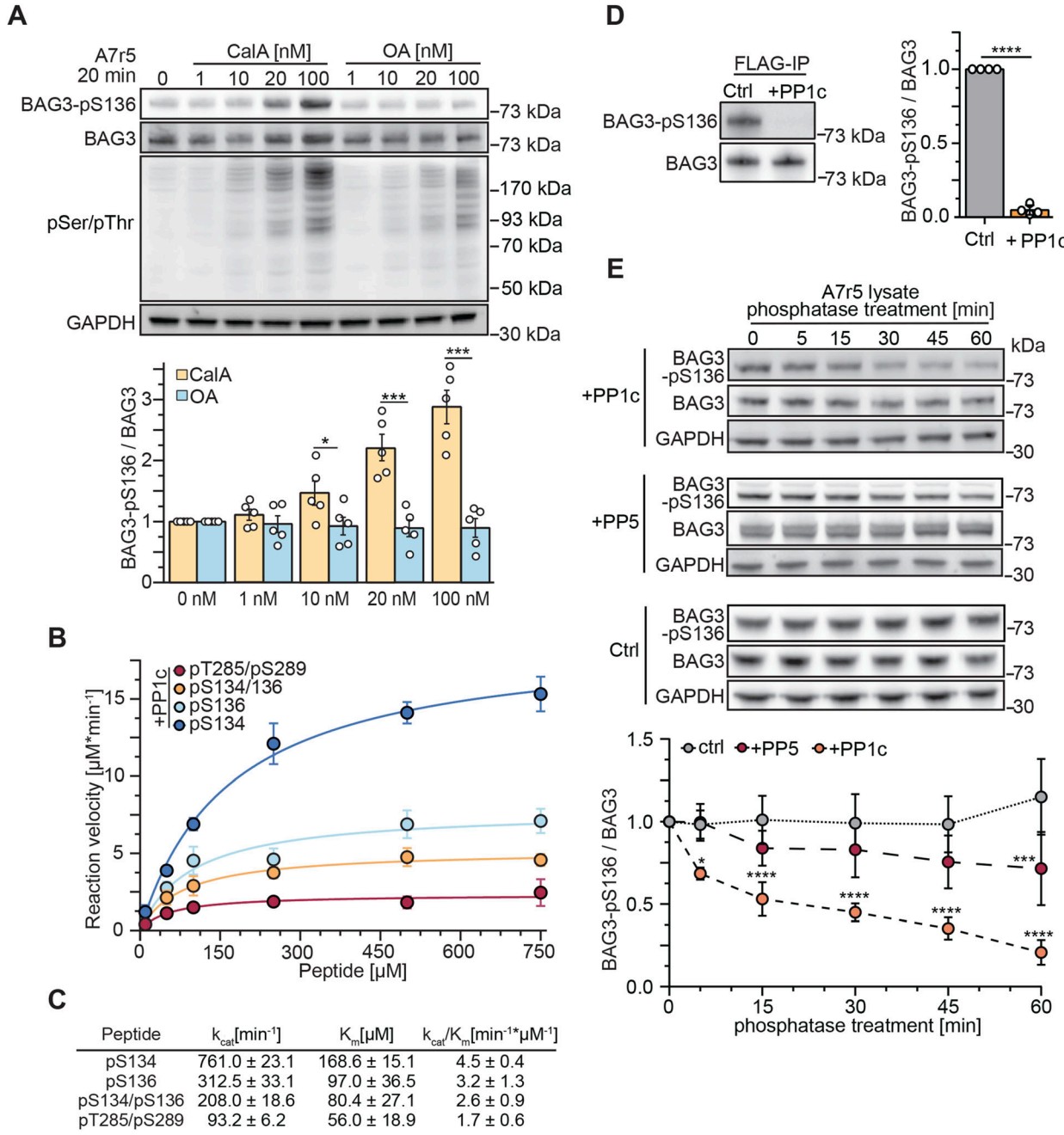

**Figure 2. PP1c dephosphorylates BAG3-pS136 in vitro.**
**(A)** A7r5 smooth muscle cells were treated with increasing concentrations of phosphoprotein phosphatase inhibitors CalA or OA as indicated, or DMSO as control. After 20 min of incubation, cells were lysed, and modulator effects on BAG3-pS136 were determined with immunoblots. The quantification is presented as a bar plot, with the mean depicted with error bars that represent the SEM based on five independent experiments. Statistical significance between inhibitors was determined with $t$ test and $P$-values are displayed as stars (*$P$ = 0.020, ***$P$ < 0.001). **(B, C)** Michaelis–Menten kinetic parameters of PP1c were determined against the indicated mono- or bisphosphorylated peptides. Error bars represent the SEM of three independent replicates with technical duplicates. $k_{cat}/K_m$ was calculated by comparison with a phosphate standard curve. **(D)** Overexpressed FLAG-BAG3 from transiently transfected HEK293 cells was single-step affinity immobilized with anti-FLAG beads and treated with PP1c for 30 min. Immunoblot quantification depicts the mean of four independent experiments, the error bar represents the SD, with $P$-values obtained from a $t$ tests (two-tailed, paired, ****$P$ < 0.001). **(E)** Monitoring of BAG3-pS136 dephosphorylation in A7r5 lysate upon incubation with PP1c, PP5, or without phosphatase as a control treatment for the indicated incubation time. Endogenous phosphatases were inhibited by the addition of 20 nM CalA. Quantification depicts the results of four independent experiments with $P$-values obtained from two-way ANOVA with Sidak correction (**$P$ = 0.006 [PP1c], **$P$ = 0.005 [PP5], ****$P$ < 0.001). Mean is shown and error bars represent the SEM.

Source data are available for this figure.

Next, to determine whether the catalytic subunit of PP1, PP1c, can directly dephosphorylate the BAG3-pS134/pS136 motif in vitro, phosphopeptides were synthesized. The peptides included the conserved amino acid (aa) stretch around the BAG3-S136 p-site, differing only in the phospho-modifications of the sites and having at least five aas at each end of the last modification to mimic the naturally bisphosphorylated sites of the RSQS motif (Fig S2B and C). Spectrophotometric kinetics demonstrated that all phosphopeptides were dephosphorylated by PP1c (Fig 2B). However, the highest catalytic efficiency ($k_{cat}/k_M$) was observed for the mono-phosphorylated peptides with 4.5 and 3.2 $min^{-1}/\mu M^{-1}$ for BAG3-pS134 and BAG3-pS136, respectively (Fig 2C). Furthermore, PP1c efficiently dephosphorylated the bisphosphorylated peptide mimicking the fully phosphorylated 14-3-3 motif (pS134/pS136), albeit with reduced efficiency compared with mono-phosphorylated peptides. To compare the dephosphorylation preference of PP1c between the bisphosphorylated 14-3-3 motif and other described BAG3 p-sites, we assessed another bisphosphorylated peptide mimicking the BAG3-pT285/pS289 sites (Fig S2B and C). The peptide containing pT285/pS289 was found to be a less favorable substrate for PP1c compared with the pS134/pS136 phosphomimicking peptide (Fig 2B and C).

We then examined PP1c's ability to dephosphorylate BAG3 at the protein level. Human FLAG-tagged BAG3 was overexpressed in HEK293 cells, affinity-enriched using FLAG beads, and subsequently exposed to recombinant PP1c on the beads. Immunoblot analysis showed nearly complete dephosphorylation of BAG3-pS136 after PP1c treatment (Fig 2D). Quantification confirmed PP1c-mediated dephosphorylation of BAG3-pS136 in a non-competitive in vitro environment. In addition, we performed dephosphorylation assays on endogenous BAG3-pS136 in A7r5 cell lysates, monitoring dephosphorylation kinetics through immunoblots. For this, A7r5 cells were lysed, and endogenous phosphatase activity was suppressed with 20 nM CalA (as shown in Fig 2A). Lysates were then treated with recombinant PP1c or PP5 at levels sufficient to overcome CalA inhibition (Fig 2E). Interestingly, PP5, the most enriched phosphatase in the BAG3 co-immunoprecipitation (co-IP) dataset, exhibited slower dephosphorylation kinetics for BAG3-pS136 compared with its known substrate p-site, pS13, on the Hsp90 co-chaperone Cdc37 (CDC37; Fig S2D) (Dushukyan et al, 2017). After 60 min of incubation, PP5 achieved significant dephosphorylation of BAG3-pS136, whereas PP1c reached a comparable dephosphorylation level within just 5 min (Fig 2E). By 60 min, PP1c had effectively dephosphorylated most of BAG3-pS136 (80%). Control samples without added recombinant phosphatase showed no change in BAG3-pS136 phosphorylation levels. Taken together, our cellular PPP-inhibitor screen and in vitro results indicate that BAG3-pS136 dephosphorylation is primarily mediated by PP1 under the tested conditions.

### PP1 dephosphorylates BAG3-pS136 in cells and mediates the interaction with 14-3-3γ

To further explore the impact of modulating endogenous PP1 activity on BAG3-pS136, HEK293 cells overexpressing BAG3 were subjected to chemical modulation of endogenous PP1 activity. On the one hand, PP1 was inhibited using Tautomycetin (TTN) and CalA (Fig 3A). TTN, renowned as the most specific PP1 inhibitor to date,

exhibits a 139-fold higher specificity towards PP1 compared with PP2A and over 300-fold higher specificity over PP5 in vitro (Choy et al, 2017). On the other hand, cells were treated with the PP1c-activating PP1 disrupting peptide (PDP) known as PDP-Nal (Fig 3A) (Wang et al, 2019). PDP-Nal, a 23-mer peptide, functions by dissociating PP1c from holoenzymes within cells, thereby enabling PP1c to dephosphorylate nearby substrates. As a control, PDPm-Nal, a variant incapable of binding to PP1c, was used (Wang et al, 2019). Immunoblot analysis was used to monitor the levels of BAG3-pS136 relative to total BAG3 in each sample upon treatment (Fig 3A). Inhibition of PP1 resulted in a 2.5-fold increase in BAG3-pS136 with TTN and a twofold increase with CalA. Complementary, treatment with PDP-Nal resulted in significant dephosphorylation of BAG3-pS136, whereas the PDPm-Nal control treatment had no effect on the p-site (Fig 3A).

To corroborate the action of PP1 on endogenous BAG3-pS136 phosphorylation, a TTN titration screen in A7r5 smooth muscle cells was carried out (Fig 3B). Immunoblot quantification revealed a correlation between increasing TTN concentration and a rising BAG3-pS136 phosphorylation (Fig 3B). Higher TTN concentrations resulted in a significant increase in phosphorylation, corroborating the regulation of the BAG3-pS136 site by PP1 in A7r5 cells. Functionally, BAG3-S136 mutants resembling the dephosphorylated state (S136A) lose their interaction with 14-3-3γ and the dynein intermediate chain (Xu et al, 2013). Consequently, increased phosphorylation of endogenous BAG3 at S136 is expected to enhance binding to 14-3-3γ, whereas PP1 activation should reduce 14-3-3γ binding. Modulation of PP1 activity was used as before (see Fig 3A) to induce varying levels of BAG3-pS136 phosphorylation. FLAG-BAG3-overexpressing HEK293 cells were treated with CalA or PDP-Nal before lysis, followed by incubation of lysates with anti-FLAG beads to capture FLAG-tagged BAG3 and its interacting proteins (Fig 3C). Immunoblot quantification revealed that PP1 inhibition led to increased BAG3-pS136 phosphorylation and enhanced binding to 14-3-3γ, whereas PP1 activation resulted in BAG3-pS136 dephosphorylation and loss of 14-3-3γ interaction (Fig 3C), illustrating a direct and rapid functional impact of PP1 activity on BAG3-pS136 in smooth muscle cells. Next, we investigated the impact of PP1 depletion on BAG3. Using pre-designed siRNAs targeting specific PP1 isoforms in HEK293 cells (Fig S3A), we observed a significant reduction in overall PP1c or PP1β levels after 48 h, resulting in a sharp increase in BAG3 levels (Fig 3D and E). However, over this extended time frame (24/48 h), no changes in BAG3-pS136 levels were observed relative to the overall BAG3 abundance (Fig S3B).

In summary, our findings indicate that PP1 regulates BAG3-pS136 in a cellular context. However, compensatory mechanisms may emerge within 24–48 h of PP1 depletion, suggesting that the accumulation of BAG3 subsequent to PP1 knockdown is not solely dependent on BAG3-pS136 phosphorylation, whereas the binding of 14-3-3γ is directly regulated through dephosphorylation by PP1 in a short time frame.

### PP5's interaction with BAG3 and chaperones supports its role in proteostasis

PP5 was found to be significantly enriched in the BAG3-associated co-IP (Fig 1B) but did not effectively dephosphorylate BAG3-pS136

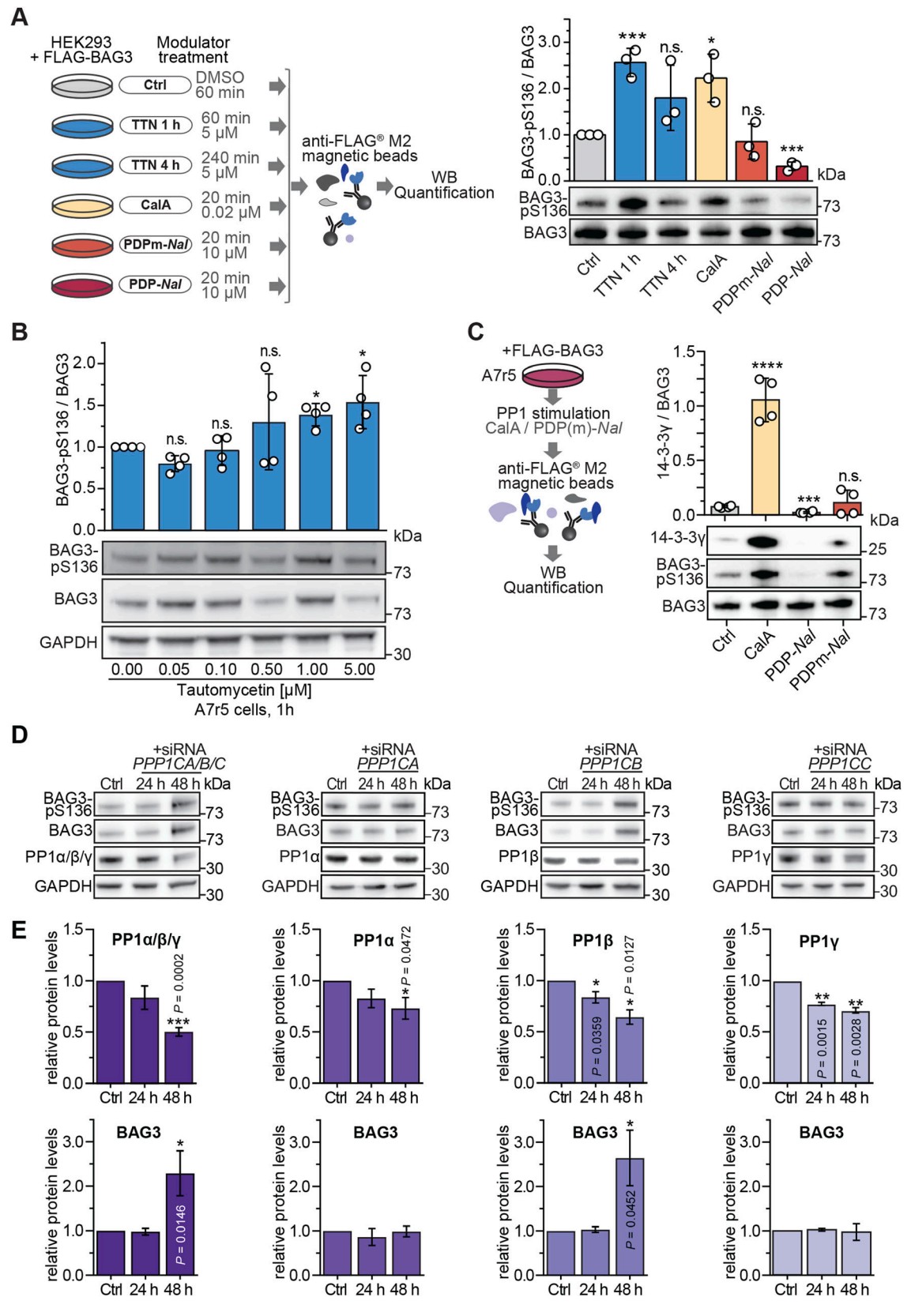

(Fig 2E). Nonetheless, PP5 is known to interact with various HSPs, which, such as BAG3, are key components of protein homeostasis networks (Connarn et al, 2014; Oberoi et al, 2022). However, compared with other PPPs, knowledge of PP5's protein interactions remains limited, with insights primarily from proteomic studies (Skarra et al, 2011) and single direct approaches (Damle & Köhn, 2019). To further explore the PP5 interactome and investigate whether co-enriched proteins are reportedly involved in proteostasis networks, FLAG-PP5 was overexpressed in HEK293 cells and subjected to AP-MS analysis (Fig S4A). Among the 236 significantly co-enriched proteins (Table S2), five reported PP5 interactors were identified (n ≥ 2 evidences in BioGRID), namely HSP90AA1 (Zeke et al, 2005), CDC37 (Skarra et al, 2011), HSP90AB1, CHORDC1, and AHSA1 (Taipale et al, 2014) (Fig 4A). In addition to these known interactors, BAG3 emerged as one of the proteins with the highest fold enrichment, reaffirming its association with PP5 (Figs 1B and 4A). GO term analysis of the enriched proteins unveiled biological processes closely linked to protein homeostasis, such as chaperone mediated protein folding, regulation of macroautophagy, protein stabilization, intracellular protein transport, and protein folding (Fig 4B). Furthermore, the comparative analysis of the enriched genes in the PP5 interactome with the entire human genome-wide expression dataset available in the PANTHER database (PANTHER17.0, reference dataset based on UniProt Release 2022_02) (Mi et al, 2019) confirmed a pronounced enrichment of proteins involved in protein folding, with a 13.5-fold increase in relation to overall gene expression (Fig 4C). To investigate whether PP5 and BAG3 contribute to the same processes or interact with the same proteins, we constructed a protein–protein interaction network (PPIN) comprising 235 co-enriched proteins using the Cytoscape plugin stringApp. The PPIN consisted of 222 nodes (proteins) and 1,237 edges (protein–protein interactions) after the removal of 13 isolated nodes (proteins without interactions) (Fig S4B). Topological analysis of the PPIN, focusing on the degree of the nodes (protein interactions of the respective protein), highlighted that PP5 and BAG3 have degrees of 22 and 25, respectively, emphasizing their potential central position and functional relevance within the PPIN. Subsequently, we used the Cytoscape plugin MCODE for clustering analysis, identifying 14 clusters labeled as C1 to C14 (Fig S4B). Notably, the genes *PPP5C* and *BAG3* formed Cluster 1 (C1) within this clustering analysis, consisting of 19 nodes and 158 edges (Fig 4D). Further String database enrichment analysis of C1 revealed that a total of eight genes, including *PPP5C* and BAG3, were significantly associated with the biological process termed "cellular response to

stress" (Fig 4D, black encirclement of nodes). These findings give insights into the related protein networks of PP5 and BAG3.

## PP5 acts as a phosphatase for a p-site cluster in BAG3

The co-IP experiments provided robust evidence of a strong association between PP5 and BAG3, supported by their concurrent presence in both datasets (Figs 1B and 4A). However, when BAG3 was subjected to in vitro treatment with PP5, we observed that BAG3-pS136 is not a suitable substrate for PP5 in vitro (Fig 2E). Moreover, co-overexpression of PP5 and BAG3 did not have an influence on BAG3-pS136 (Fig 5A). Alongside BAG3-pS136, the p-sites pS284, pT285, pS289, and pS291 collectively constitute a p-site cluster in BAG3 and were shown to have functional significance (Fig 1A) (Kim et al, 2016; Zhou et al, 2020; Luthold et al, 2021; Ottensmeyer et al, 2024). The BAG3 multiple sequence alignment revealed the conservation of this cluster across mammalian species (Fig S1C). Consequently, our next objective was to determine whether PP5 is the phosphatase responsible for dephosphorylating this cluster.

First, we evaluated whether BAG3-pT285 and -pS289 mimicking phosphopeptides could be dephosphorylated with PP5c, which consists solely of the catalytic subunit and lacks its autoinhibitory TPR domain. To assess dephosphorylation kinetics, peptide-based phosphatase assays were carried out, representing either BAG3-pT285 or BAG3-pS289, or both as a bisphosphorylated peptide (pT285/pS289) (Figs S2C and 5B). The spectrophotometric kinetics analysis revealed the highest catalytic efficiency of 2.8 $k_{cat}/K_M$ for PP5c towards the pT285 peptide compared with 1.1 and 1.2 for pS289 and pT285/pS289, respectively (Fig 5B). In addition, it is visible that PP5c dephosphorylated the 14-3-3 motif mimicking peptide (pS134/pS136) slower, thereby providing additional evidence to support its unsuitability as a substrate for PP5. Next, the impact of PP5 on the BAG3 p-site cluster through both in vitro and in cells was assessed on the protein scale. HEK293 cells overexpressing FLAG-tagged BAG3 were subjected to treatment with recombinant PP5 post-lysis (in vitro), or transiently co-transfected with PP5 (in-cell) or remained untreated (Ctrl) (Fig S5A). Upon lysis, endogenous phosphatases were inhibited by the addition of 20 nM CalA. The samples were analyzed using parallel reaction monitoring-based targeted phosphoproteomics and normalized the ratios using indexed retention time (iRT). The resulting peptides of the trypsinization (SSTPLHSPSPIR) were found to be phosphorylated at multiple sites in various combinations (Fig 5C). Recombinant PP5 effectively dephosphorylated singly and doubly phosphorylated

---

**Figure 3. PP1 dephosphorylates BAG3-pS136 in cells.**
**(A)** Illustrative overview of the small molecule and peptide modulator screen to tune PP1 activity in cells. HEK293 cells transiently expressing FLAG-BAG3 were treated as illustrated, lysed, and captured by a single-step affinity enrichment using anti-FLAG beads. Samples were analyzed by immunoblots. Quantification depicts results of three independent experiments with P-values obtained from t tests (two-tailed). Mean is shown with replicates as scatter plot and error bars represent the SD (*P = 0.015, ***P < 0.001). **(B)** A7r5 smooth muscle cells were incubated with increasing concentrations of Tautomycetin as indicated. Immunoblots of lysate were used to determine the sensitivity of the titrated inhibitor towards BAG3-pS136. Mean is shown as a barplot with replicates as scatter plots, and error bars represent the SD of four independent experiments. Statistical significance between concentrations is determined with t test with Welch's correction (*P < 0.05). **(C)** Analysis of phosphorylation-dependent binding of 14-3-3γ to FLAG-BAG3 in A7r5 muscle cells after 20-min modulation of endogenous PP1 before lysis and FLAG-IP enrichment. Results are presented as a barplot, P-values obtained from a t test (***P = 0.0002, ****P < 0.0001). **(D)** Immunoblots depicting siRNA-mediated knockdown of all PP1 isoforms combined (PP1α/β/γ) or individual PP1 isoforms (PP1α, PP1β, PP1γ) separately. The displayed blots represent the analyzed protein levels of 24 or 48 h after transfection. **(E)** Quantification of immunoblotted lysates show level changes of PP1α/β/γ and BAG3 (n = 3 or 4). Quantification depicts results of three or four independent experiments with P-values obtained from t tests (two-tailed, paired). Error bars represent the SD.
Source data are available for this figure.

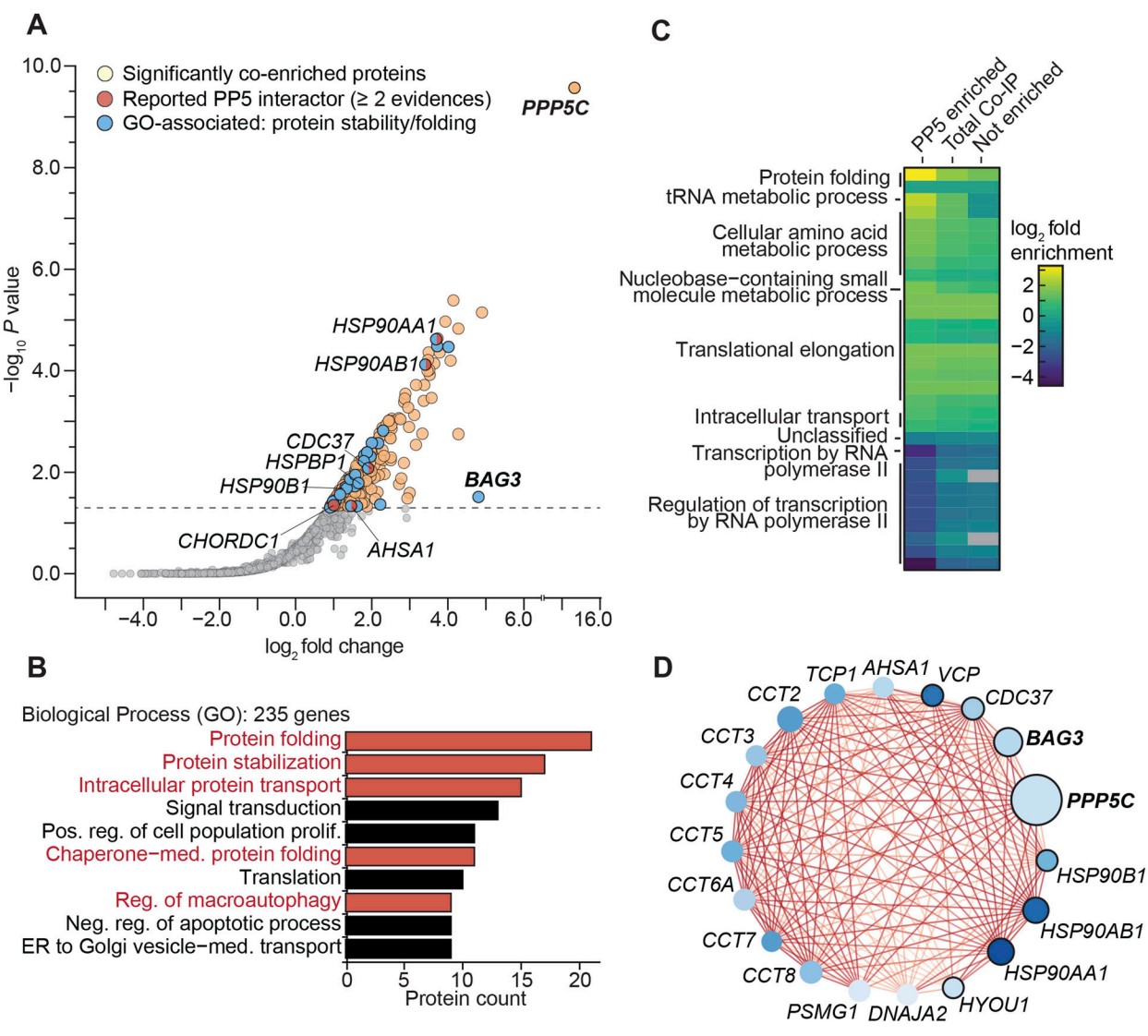

**Figure 4. PP5 is part of the protein maintenance network.**
**(A)** Co-IP of FLAG-PP5 from HEK293 cells with significantly enriched proteins highlighted. Enriched proteins in the sample are marked in the respective colors for the subgroups. Enrichment was determined through rankprod (n = 4). Selected hits are annotated with their gene name. Gene names of proteins of interest in this work are highlighted in bold. Dashed line indicates a false discovery rate of 5%. **(A, B)** Gene ontology analysis of the significantly enriched genes from the PP5 co-IP sample (n = 235) from (A). The 10 most abundant biological process GO terms are shown as bar charts, ranked based on the protein counts. Biological processes related to protein folding and stabilization are illustrated in red. **(A, C)** Enrichment analysis of enriched genes from the co-IP experiment (A) compared with overall human gene expression displayed as a heatmap. Fold enrichment was calculated for the enriched, not enriched, and total sample, with overall GO categories annotated at the respective cluster. Statistical significance was determined using Fisher's test (P-value < 0.05) with the Bonferroni correction implemented in the PANTHER database. Non-significant changes are colored as gray within the heatmap. **(D)** Cluster 1 of PP5 co-enriched proteins, with key characteristics visually represented through color-coding and labeled with their respective gene name. Proteins are presented as nodes: darker blue hues signify more interactions within the cluster. Node size reflects the fold change in the co-IP, with larger nodes representing higher fold change values. Black encirclement of nodes indicates the GO biological process "cellular response to stress." Connecting lines (edges) between nodes display interactions; the color depicted the confidence score, with darker lines indicating higher confidence scores.

sites including pS284/pT285/pS289/pS291 in different combinations (Fig 5C). Considering all results, this demonstrates a differentiation of PP5-mediated dephosphorylation of BAG3 p-sites (Figs 2E and 5A–C). Notably, the addition of arachidonic acid, a small molecule activator of PP5 (Chen & Cohen, 1997), did not induce BAG3 dephosphorylation, possibly hindering the necessary interaction between PP5's TPR domain with its binding protein (Fig S5B). After confirming that PP5 dephosphorylates the BAG3 p-site cluster both

in vitro and in cells, we inquired whether phosphorylation of BAG3 in this cluster would regulate PP5 binding to BAG3. To this end, we co-precipitated overexpressed FLAG-tagged WT, S284A/T285A/S289A/S291A (4A) and S284D/T285D/S289D/S291D (4D) BAG3 variants and measured PP5 binding by MS read out (Fig 5D). We observed that BAG3_4D bound significantly stronger to PP5 than BAG3_4A, with the wt variant—being in part phosphorylated (Fig 5C)—binding to PP5 with an intermediate efficacy, suggesting that

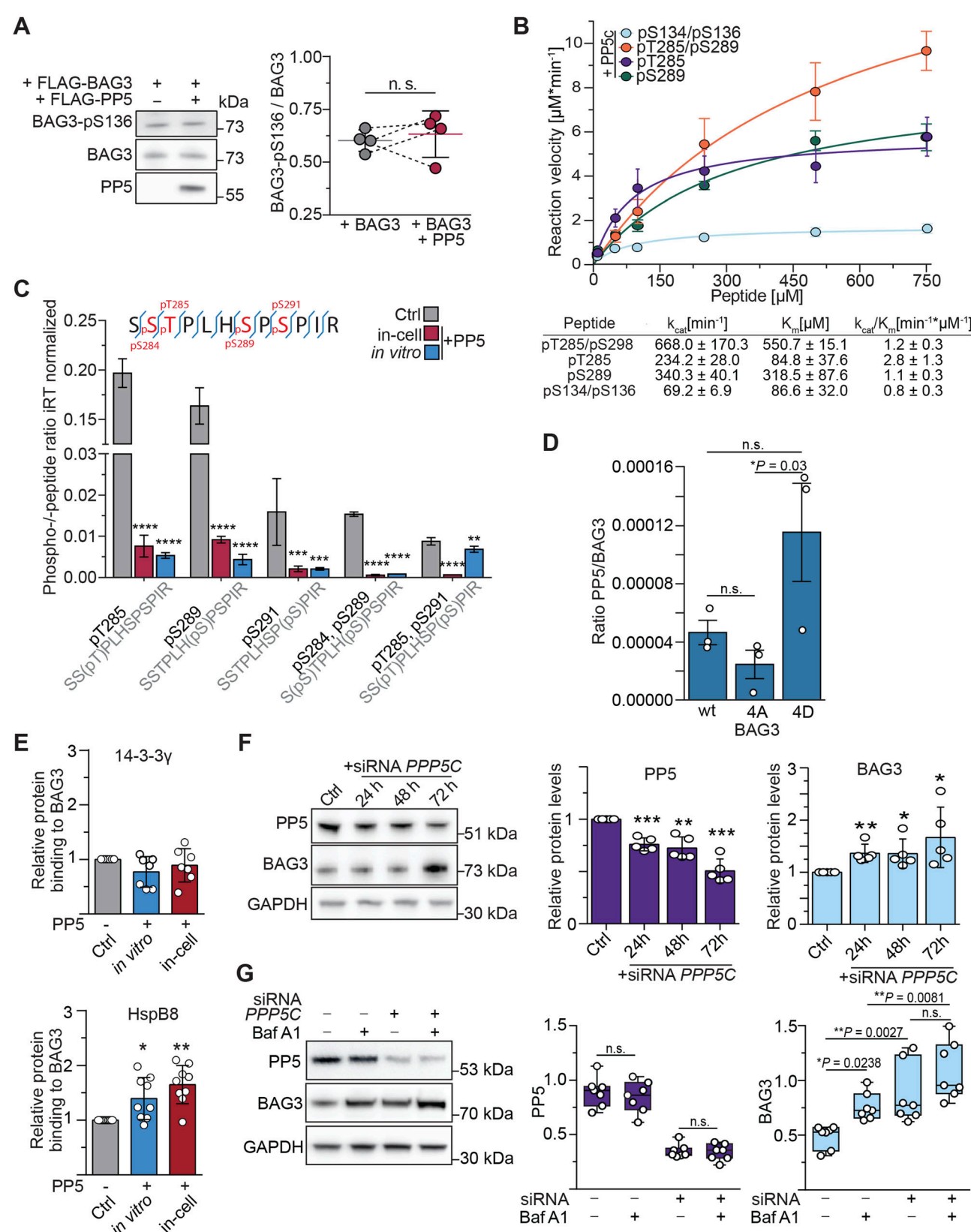

**Figure 5. PP5 dephosphorylates a p-site cluster in BAG3 and enables HspB8 binding.**
**(A)** Phosphorylation levels of BAG3-pS136 were compared in HEK293 cells overexpressing FLAG-BAG3 or both FLAG-BAG3 and FLAG-PP5 for 24 h. Immunoblot quantification demonstrates that PP5 overexpression does not affect BAG3-pS136 phosphorylation. Statistical significance was determined through *t* test (n = 4).

the negative charges introduced by phosphate groups support PP5 binding. We then used the same treatment as for the experiment in Fig 5C with an immunoblot readout to explore changes in protein interactions with dephosphorylated BAG3. We observed that dephosphorylation of the BAG3 p-site cluster significantly enhanced HspB8 binding to BAG3 in vitro and in cells (Figs 5E and S5C and D). As expected, binding of 14-3-3γ upon PP5-mediated dephosphorylation was unchanged (Figs 5E and S5C and D). To evaluate the effect of PP5 depletion on BAG3 protein levels, A7r5 cells underwent PPP5C siRNA transfection. The knockdown of PPP5C resulted in a notable reduction in PP5 levels after 24 h, as determined by immunoblot analysis (Fig 5F). Concurrently, there was a significant increase in BAG3 levels (Fig 5F). Quantitative analysis of the knockdown revealed a strong negative correlation between the protein levels of PP5 and BAG3, as evidenced by the Spearman correlation (Fig S5E). Notably, BAG3 is co-degraded during CASA, when autophagic membranes enclose the BAG3-chaperone complex followed by autophagosome-lysosome fusion (Arndt et al, 2010; Ulbricht & Höhfeld, 2013; Ottensmeyer et al, 2024). The observed increase in BAG3 levels upon PP5 depletion may thus reflect inhibition of the CASA pathway. To further verify this, CASA was induced in A7r5 cells through cultivation on fibronectin-coated plates (Ulbricht & Höhfeld, 2013) and treated with (±) siPP5 in the presence or absence of (±) Bafilomycin A1 (BafA1), an inhibitor of lysosomal acidification that blocks the later stages of autophagy (Fig 5G) (Ottensmeyer et al, 2024). BafA1 treatment alone did not affect PP5 levels, but BAG3 levels increased comparably upon both CASA inhibition and PP5 depletion. Notably, the combination of PP5 depletion and CASA inhibition resulted in the most pronounced BAG3 accumulation (Fig 5G). In addition, we assessed the protein levels of Sequestosome-1 (SQSTM1), a recently reported kinase regulator of pT285 (Luthold et al, 2021) and a component of the CASA pathway. SQSTM1 exhibited a similar enrichment pattern upon PP5 depletion and CASA inhibition, mirroring the results for BAG3 (Fig S5F).

Taken together, we demonstrated that PP5 can dephosphorylate the BAG3 p-site cluster including pS284, pT285, pS289, and pS291 in vitro and in cells, facilitating the binding of HspB8 to dephosphorylated BAG3. Moreover, depletion of PP5 led to the accumulation of BAG3 and SQSTM1 in A7r5 smooth muscle cells, pointing to a role of the phosphatase in CASA activation in smooth muscle cells.

# Discussion

Here, we describe that PP1 and PP5 can dephosphorylate BAG3, thereby affecting its protein–protein interaction and proteostasis functions. Assigning specific kinases and phosphatases to distinct p-sites is inherently challenging (Needham et al, 2019). To date, no phosphatase and only two kinases (CDK1, PKCδ) have been identified for BAG3 phosphorylation in cells (Zhou et al, 2020; Luthold et al, 2021). Kinase motifs and prediction tools often struggle with closely clustered phosphorylation sites, typically suggesting a broad family of kinases rather than a single candidate (Johnson et al, 2023). The complexity in enzyme assignment stems from the dynamic behavior of phosphorylated residues and the multifaceted regulation of both kinases and phosphatases (Bollen et al, 2010; Miller & Turk, 2018; Nilsson, 2019; Hoermann et al, 2020; Kruse et al, 2020; Kokot & Köhn, 2022; Nguyen & Kettenbach, 2023). As a result, a single assay is typically inadequate for definitively identifying the phosphatase responsible for the dephosphorylation of a particular residue (Fahs et al, 2016; Zhang et al, 2021). As a starting point to identify potential phosphatase candidates for BAG3, we carried out a co-IP analysis of BAG3. Previously reported interactors were successfully enriched, supporting the validity of the approach. Importantly, the co-IP of BAG3 identified phosphatases and several PP1 and PP2A regulatory subunits, which can "scaffold" the respective phosphatases and their substrate (Fahs et al, 2016). As a result, in such studies, catalytic subunits are less frequently observed than regulatory subunits (Fahs et al, 2016; Kokot & Köhn, 2022).

Because of its functional relevance, we first focused on BAG3-pS136 dephosphorylation and dissected PPP activities using inhibitors, which revealed PP1 as likely to be involved. Nevertheless, a clear selectivity is challenging to achieve with inhibitors (Swingle et al, 2007). Among the inhibitors applied here, TTN was previously shown to be selective in vitro (Choy et al, 2017). TTN treatment in cells requires longer incubation times and higher concentrations than in vitro, resulting in a lesser specificity than observed in vitro, but it remains the most specific inhibitor available (Mitsuhashi et al, 2003; Huang et al, 2014; Choy et al, 2017; Messal et al, 2019; Zhang et al, 2021). In addition, because reversible phosphorylation

---

**(B)** Michaelis–Menten kinetic parameters of PP5c were determined against the indicated mono- or bisphosphorylated peptides. Error bars represent the SEM of three independent repeats with technical duplicates. $k_{cat}/K_m$ was calculated by comparison with a phosphate standard curve. **(C)** HEK293 cells overexpressed FLAG-BAG3 and were treated with 1 μM recombinant PP5 after lysis for 1 h (in vitro), or overexpressed FLAG-BAG3 and FLAG-PP5 simultaneously for 24 h (in-cell), or were not treated with phosphatase (Ctrl). FLAG-BAG3 was subjected to affinity enrichment and subsequently evaluated using LC–MS/MS. The theoretical phosphopeptide generated through trypsinization is shown at the top. Targeted phosphoproteomics quantification reveals the complete dephosphorylation of the p-site cluster for both PP5 treatments. Statistical significance was determined through two-way ANOVA (n = 4), error bars represent the SD (****$P < 0.0001$, ***$P = 0.0002$, *$P = 0.0021$). Indexed retention time (iRT). **(D)** FLAG-BAG3 mutants mimicking the phosphorylated (4D) and dephosphorylated (4A) state of the BAG3 p-site cluster were overexpressed in HEK293 cells followed by co-IP and MS analysis to assess PP5 binding to each variant compared with WT. Bars represent the mean (n = 3), with significance and error bars calculated using t test (*$P = 0.03$). **(C, E)** Samples were subjected to the same treatment as described for (C) to induce dephosphorylation of p-site cluster. This was followed by the analysis of protein quantities bound to BAG3 via immunoblotting, normalized to the amounts bound to the control sample. Statistical significance of changes in protein levels was assessed using data from seven or nine independent replicates, with means presented as bar plots. $P$-values were calculated using a one-way ANOVA with a 95% confidence interval and analyzed using PRISM/GraphPad version 6 (HspB8: *$P = 0.0188$; **$P = 0.0002$; 14-3-3: $P$ [in vitro] = 0.1668, $P$ [in-cell] = 0.6241). **(F)** A7r5 cells were transfected with PPP5C siRNA for the specified incubation time and protein levels were determined using immunoblots. Statistical significance in the changes of protein levels was determined based on data from five independent replicates, and the means are presented as bar plots. The $P$-value was calculated using a two-tailed test with a 95% confidence interval and determined using PRISM/GraphPad version 6 (PP5: ***$P = 0.001$, **$P = 0.0045$, ***$P = 0.0006$; BAG3: **$P = 0.0097$, *$P = 0.046$, *$P = 0.0328$). **(G)** Smooth muscle cells were subjected to treatment with control siRNA or PP5-targeting siRNA (siPP5) followed by treatment with BafA1 to inhibit CASA mediated BAG3 degradation. Statistical significance of changes in protein levels was assessed using data from seven independent replicates, represented as boxplots. $P$-values were calculated using a two-way ANOVA and analyzed using PRISM/GraphPad version 6.
Source data are available for this figure.

signaling occurs rapidly (Reddy et al, 2016), extended incubation times may lead to indirect or diminishing effects as observed here when cells were incubated for 4 h with TTN (Fig 3A). We therefore assessed the effects of both inhibiting and activating endogenous PP1 activity on BAG3-pS136 phosphorylation using various modulators with short incubation times. Our findings showed that inhibiting PP1 increased BAG3-pS136 phosphorylation, whereas activating PP1 led to significant dephosphorylation (Figs 2A and 3A and C). This demonstrates that PP1 dephosphorylates BAG3-pS136 in cells, a finding that is supported by experiments showing direct dephosphorylation in vitro (Fig 2B–E). In addition to the possibility of interacting with BAG3 through regulatory subunits, BAG3 may interact with PP1 directly in a transient manner not easily detectable by IPs. If so, identifying the binding site is challenging because of the absence of a typical binding motif, such as the RVxF motif, which could provide clues (Wakula et al, 2003; Bollen et al, 2010). However, this does not rule out BAG3 as a PP1 interactor, as some proteins, such as SDS22, bind PP1 without such motifs (Bollen et al, 2010; Heroes et al, 2019). Therefore, the complex question of how exactly PP1 interacts with its substrate BAG3 will be part of future studies.

In terms of functional consequences, connecting rapid signaling events such as (de-) phosphorylation (Reddy et al, 2016) to changes in protein levels, which occur over multiple hours (Chen et al, 2016), remains a challenge. This temporal discrepancy of cause and effect further complicates the overall difficult assignment of PPP holoenzymes to specific sites (Fahs et al, 2016). In addition, whereas PP1 knockdown offers a selectivity that cannot be achieved with inhibitors, it is also known to cause pleiotropic effects (Fahs et al, 2016). As a result, we were not able to establish a direct connection between PP1c knockdown, BAG3-pS136 phosphorylation, and BAG3 protein levels (Figs 3D and E and S3B). However, when focusing on the immediate effects of PP1 activation or inhibition in the cell, we observed that the activity of PP1 directly affected the binding of 14-3-3γ to BAG3 via pS136 dephosphorylation (Fig 3C). This phosphorylation-dependent interaction, previously investigated with mutants (S136A/S136D), was shown to regulate the translocation of protein aggregates to the aggresome and modulate CASA activity (Xu et al, 2013; Ottensmeyer et al, 2024).

Whereas PP5 dephosphorylated BAG3 at pS136 only to a very limited extent, it emerged as a strong phosphatase candidate for BAG3 because of their consistent co-occurrence in co-IPs (Figs 1B and 4A). Interestingly, we did not observe the co-occurrence of HSP70/HSP90, well-known activators and substrate-directing proteins of PP5, in both co-immunoprecipitations. However, a recent study demonstrates that several other proteins can activate PP5 without the involvement of HSPs (Devi et al, 2024). Analysis of the PP5 interaction network revealed proteostasis-related GO biological processes (Fig 4C and D), further linking PP5 to BAG3-involved pathways (Klimek et al, 2017; Höhfeld et al, 2021; Kirk et al, 2021).

Residues of this p-site cluster are dephosphorylated in response to mechanical forces in muscle cells (Hoffman et al, 2015; Potts et al, 2017; Ottensmeyer et al, 2024). In the targeted phosphoproteomics approaches we then observed the dephosphorylation by PP5 of a whole BAG3 p-site cluster, which is conserved in mammals (Figs S1B and 5C). Our results are in line with recent studies demonstrating

that PP5, when bound in a complex, displays some degree of freedom of the liberated PP5c domain to dephosphorylate p-site clusters of bound phosphoproteins (Oberoi et al, 2022; Jaime-Garza et al, 2023). In contrast, PP5 did not dephosphorylate BAG3-pS136 in cells, indicating a differentiation between the p-sites under the same conditions (Fig 5B). This could be caused by a spatial distance between the PP5 active site and BAG3-pS136 or a negative substrate selectivity as observed in the phosphopeptide and in vitro assays (Figs 2B and E and 5B), or a combination of both. Recent studies have unveiled negative selectivity as a mechanism guiding substrate specificity for serine/threonine kinases and phosphatases (Hoermann et al, 2020; Hein et al, 2023; Johnson et al, 2023).

Phosphorylation-dead mutants of BAG3-pT285 and -pS289 (T/S to A) enabled BAG3 to bind to HspB8 (Ottensmeyer et al, 2024). Our findings further show that PP5-mediated dephosphorylation of the BAG3 p-site cluster (pS284, pT285, pS289, pS291) in HEK293 cells increases HspB8 binding (Fig 5E). These findings introduce an additional layer of phosphorylation-mediated regulation of BAG3 PPIs, showing that HspB8 binding is controlled by phosphorylation sites distal to BAG3's isoleucine-proline-valine (IPV) motifs (amino acids 96–98 and 208–210) that mediate the binding of HSPs (Fuchs et al, 2009). Furthermore, our results suggest that PP5 could serve as a physiological antagonist of some CDKs that phosphorylate this p-site cluster (Zhou et al, 2020; Luthold et al, 2021). However, phosphorylation of S291 by CDK5 destabilized BAG3 in neurons (Zhou et al, 2020; Luthold et al, 2021), whereas in smooth muscle cells we observed stabilization of BAG3 protein levels upon PP5 depletion correlating with the dephosphorylation of the p-site cluster upon PP5 overexpression (Fig 5C, F, and G). This could suggest that the phosphorylation status leads to different consequences in different cell types, or that the effect on BAG3 protein levels depends on the phosphorylation pattern of those p-sites. Nevertheless, our data unequivocally demonstrate that PP5 can dephosphorylate T285 and S289, which control the association of BAG3 with small heat shock proteins (Ottensmeyer et al, 2024). The increase in SQSTM1 levels, which is also involved in the phosphorylation of BAG3-pT285 in HeLa cells (Luthold et al, 2021), after PP5 depletion also suggests broader regulatory functions of PP5 (Fig S5F). Together with evidence of PP5-HSP interactions (Sinclair et al, 1999; Yang et al, 2005; Golden et al, 2008) and BAG3-HSP interactions, our data indicate a central role for PP5 in the assembly of the CASA complex (Fig 4D).

The multifaceted roles of BAG3 and its multiple transient, pathway-specific PPIs make it challenging to gain a comprehensive understanding of this protein. Previous efforts to map the BAG3 interactome have been carried out (Chen et al, 2013; Hiebel et al, 2020), but the understanding of regulatory interactions remains limited. Identifying BAG3 regulators is crucial for a deeper understanding of BAG3 pathway-specific functions and for enabling the precise control of BAG3 function by governing BAG3's PPIs in the context of diseases (Lin et al, 2022). Overall, the present study reveals a direct involvement of PP1 and PP5 phosphatases in the regulation of BAG3 interactions and BAG3 proteostasis functions (Fig 6). Our BAG3 co-IP also identified other kinase (Table S1) and phosphatase candidates, such as PP2A, PPM1A, and PPM1B (Table S1, Fig 1C). BAG3 contains multiple p-sites, and it is also possible that different phosphatases and kinases can act, depending on the

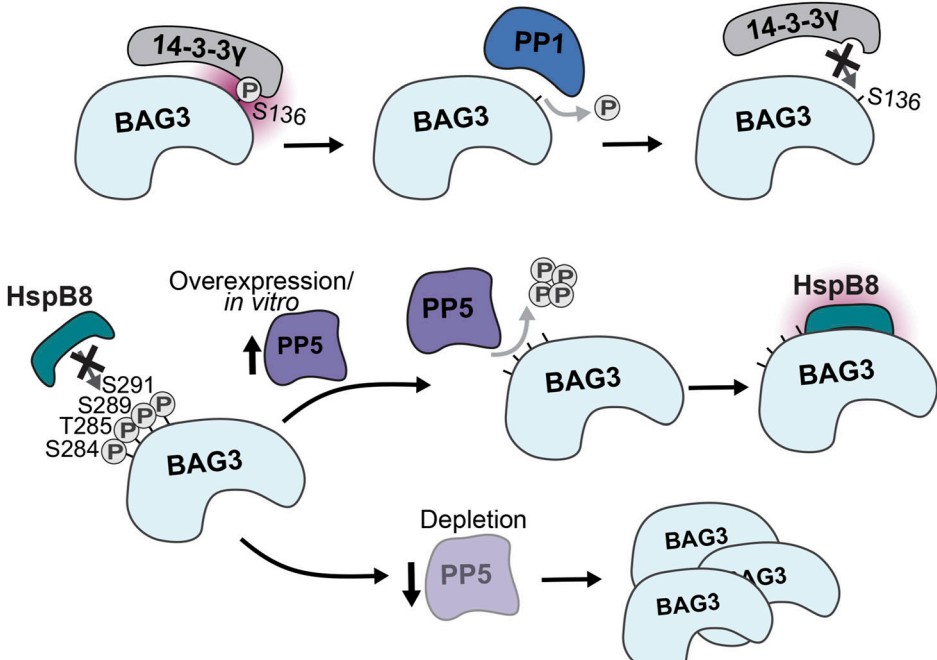

**Figure 6. Regulation of BAG3 protein interactions through PP1 and PP5.**
Dephosphorylation of pS136 on BAG3 by PP1 leads to loss of 14-3-3 protein binding. Dephosphorylation of the p-site cluster pS284-pS291 by PP5 enables HspB8 binding, and PP5 depletion increases BAG3 levels in a CASA-dependent manner.

cellular context and process, on the same p-sites. Therefore, our data also provide further phosphatase and kinase candidates to act on BAG3 in different processes to follow up on in the future.

# Materials and Methods

### Materials for phosphopeptide synthesis

Synthetic reagents were obtained from Novabiochem, Sigma-Aldrich, or Carl Roth. All amino acids and resins used in the study were obtained from Novabiochem. Peptide synthesis was conducted using a MultiPep RSi peptide synthesizer (Intavis Bioanalytical Instruments AG). The resulting peptides were purified using a 1260 Infinity II Prep HPLC system (Agilent Technologies) with a VP 250/10 NUCLEODUR 100-5 C18 ec column (Macherey-Nagel). The HPLC system used a general gradient of 10–50% acetonitrile (ACN) in $H_2O$. The peptide products were verified using a 1260 Infinity I HPLC System coupled to a 6120 Quadrupole LC/MS (Agilent Technologies) with an EC 250/4 NUCLEODUR 100-5 C18 ec column (Macherey-Nagel) running a general gradient of 10–50% ACN in $H_2O$. The peptide products were also analyzed using a Microflex LT MALDI (Bruker).

### Phosphopeptide synthesis and purification

Peptides were synthesized after fluorenylmethyl-oxycarbonyl (Fmoc) solid-phase peptide synthesis (SPPS). The peptides were synthesized on Wang resin or pre-coupled H-Lys(Boc)-2-ClTrt resin. Before synthesis, the resins were swollen in N,N-dimethylformamide (DMF) for 20 min. Each round of synthesis involved double coupling of Fmoc- and side-chain-protected

amino acids to the resin, followed by capping with acetic anhydride ($Ac_2O$) and removal of the Fmoc group with piperidine. The coupling reactions were carried out by adding Fmoc-protected amino acids (4 eq.), O-benzotriazole-N,N,N′,N′-tetramethyl-uronium hexafluorophosphate (HBTU) (4 eq.), N-hydroxybenzo-triazole hydrate (HOBt) (4 eq.), and N-methylmorpholine (NMM) to the resin in 1 ml DMF and reacting for 30–45 min. Capping was performed by adding a solution of 5% acetic anhydride $Ac_2O$ and 5% 2,6-Lutidine in DMF for 5 min. After the final amino acid residue was added, the Fmoc group was removed using a 20% piperidine in DMF solution for 3 and 8 min, respectively. Resins were washed with DMF between each step. To fully deprotect the peptides and libraries, they were removed from the resin and shaken for 3–4 h in a cleavage cocktail consisting of 92.5% trifluoroacetic acid (TFA), 5% triisopropylsilane (TIPS), and 2.5% $H_2O$. Peptides were precipitated in cold diethylether (–20°C) and collected by centrifugation. Peptides were then validated by MALDI and HPLC-MS, and purified by preparative HPLC. Phosphopeptides that were not synthesized by us were purchased from Biomatik upon a custom request.

### Spectrophotometric kinetic assays

Purified peptides were dissolved in a mixture of 10% DMSO and 90% $H_2O$ to obtain a concentration of 10 mM. To assess the kinetics/dynamics of peptide dephosphorylation by recombinant PP1c or PP5c, the EnzChek Phosphate Assay Kit from Thermo Fisher Scientific was used. The assay was conducted in a reaction buffer of 20 mM Tris, 100 mM NaCl, 2 mM DTT, 0.15 U purine nucleoside phosphorylase, 0.2 mM 2-amino-6-mercapto-7-methylpurine riboside, and various concentrations of peptides (as indicated), and 25 nM of recombinant PP1c or PP5c. The absorbance change was monitored at 360 nm at 28°C using a Synergy H1 microplate

reader (BioTek). The raw data obtained were analyzed using GraphPad Prism v6.0. The error bars represent the SD of three biological replicates conducted as two technical replicates, and the data were compared with a standard curve prepared using the phosphate solution included in the kit and following the manufacturer's instructions. The Michaelis–Menten model from Graph-Pad Prism v6.0 was used to fit the data from three independent replicates, and the kinetic parameters were extracted.

## Plasmids and cloning

The recommended protocol for polymerase chain reaction (PCR) setups from the manufacturer was followed using Phusion Polymerase (Thermo Fisher Scientific) for all clonings, whereas Fast-Digest enzymes (Thermo Fisher Scientific) were used as the restriction enzymes. Coding sequence for human PP5 was designed with NdeI/XhoI restriction sites attached at the 5' and 3' ends, respectively, which was then ordered into the pET15b plasmid through GenScript Biotech. The NEBuilderkit (NEB) was used with 5'-cgcgcggcagccatatgagc-3' (fwd) and 5'-ctcagcttcctttcgggctttgttag-3' (rev) primers to generate pET15b-PP5c from pET15b-PP5. The pCMV-3Tag-A1 plasmid was obtained from Addgene and used to insert the PP5 coding sequence, cut out using NdeI/XhoI restriction enzymes, via molecular cloning. In addition, the pCMV-3Tag-A1-BAG3-BAG3 plasmid originated from the Höhfeld laboratory and BAG3 coding sequence was cloned into pCMV-3Tag-A1 with BamHI/EcoRI to generate pCMV-3Tag-A1-BAG3. All plasmids were sequenced via Sanger sequencing to confirm their sequence integrity before further use. The BAG3 phospho-mutants were generated following the instructions of the manufacturer's QuickChange kit (Nr. 10518; Agilent). The template was the pCMV-3Tag-A1-BAG3 vector. The phospho-mimetic version (BAG3-S284D/T285D/S289D/S291D, named BAG3-4D) was generated with the primers 5'-gggctcaccagccaggagcgacgatccactccacgaccccgatcccatccgtgtgcaca ccg-3' (fwd) and 5'-cggtgtgcacacggatgggatcggggtcgtggagtggatcgtcg ctcctggctggtgagccc-3' (rev). The BAG3 dephosphorylation-mimetic BAG3-S284A/T285A/S289A/S291A (named BAG3-4A) mutant was generated through two consecutive PCRs with 5'-caccagccag-gagcgccgcgccactccactcc-3' (fwd) and 5'-ggagtggagtggcgcggcgctcct ggctggtg-3' (rev) followed by 5'-gccactccacgcccccgcgcccatccg-3' (fwd) and 5'-cggatgggcgcggggcgtggagtggc-3' (rev). The PCR product was cloned into the template to replace BAG3wt. Successful generation of BAG3 mutagenesis was confirmed through sequencing.

## Recombinant protein expression

A new batch of recombinant *PPP1CA* was expressed and purified following the established protocol (Salvi et al, 2018). The pTXB1-His-TEV-PP1α-intein protein was produced in *E. coli* BL21Star (DE3) pRARE in LB broth supplemented with 1 mM MnCl$_2$. The protein expression was induced with 50 $\mu$M isopropyl $\beta$-d-thiogalactoside (IPTG) at 16°C overnight. The wet pellet was then lysed using sonication in 25 mM TRIS–Cl with a pH of 7.5 at 4°C, and the addition of 300 mM NaCl, 10% vol/vol glycerol, 30 mM imidazole, 0.2% vol/vol tween-20, 0.1 mM phenylmethylsulfonyl fluoride, EDTA-free protease inhibitor cocktail (Roche), and benzonase (Merck). Next, the

soluble protein fraction was supplemented with NaCl to reach a final concentration of 700 mM. The protein was then purified using a HisTRAP HP nickel column (GEHealthcare) in 25 mM TRIS–HCl, pH 7.5 at 4°C, with 700 mM NaCl, 5% vol/vol glycerol, and 30 mM imidazole, followed by dilution and incubation with 50 mM $\beta$-mercaptoethanol ($\beta$-ME) and 1 mg TEV protease at 4°C overnight to cleave the tag. The cleaved PP1c was subsequently purified using chitin resin and a Heparin HP column, which was equilibrated in 20 mM TRIS–Cl at RT, with a pH of 7.5, 100 mM NaCl, and 5 mM $\beta$-ME. Elution was performed using a high salt gradient, and the protein was then dialyzed into storage buffer. pET15b-His-PP5 and pET15b-His-PP5c were used to expressed PP5 and PP5c, respectively in *E. coli* BL21Star (DE3). We followed the expression and initial expression procedure as described for PP1c. After the HisTRAP HP nickel column, we performed a dialysis o/n and continued with the Heparin HP column. Fractions containing His-PP5 and His-PP5c were pooled and dialysed o/n with storage buffer, concentrated and supplemented with glycerol to a final concentration of 20% for long-term storage at –80°C.

## Cell culture

HEK293 cells were obtained from the EMBL Heidelberg cell line repository. A7r5 rat smooth muscle cells (Sigma-Aldrich) were cultured at 37°C and 5% $CO_2$ in a humidified incubator. The cells were grown in GlutaMax DMEM (Gibco) supplied with 10% (vol/vol) FBS, 1% (vol/vol) penicillin/streptomycin, and 1 g/liter or 4.5 glucose for A7r5 or HEK293 cells, respectively. Cells were passaged routinely when reaching 90% confluency by trypsinization and 10-fold dilution.

## Small molecules and peptide modulators

Calyculin A was obtained from Cell Signaling Technologies (#9902 S; Cell Signaling Technology, Le), Okadaic acid from Merck (Merck KGaA), Tautomycetin from tocris (Cat.nr. 2305; tocris), and arachidonic acid from TCIChemicals (A0781). PDP-*Nal* and the control peptide PDPm-*Nal* were synthesized from JPT peptide technologies GmbH upon customized request. All modulators were dissolved in DMSO. Cells were incubated with 20 nM CalA or OA for 20 min, 0.05–5 $\mu$M TTN for 60–240 min, 250 $\mu$M arachidonic acid for 2 h, and 10 $\mu$M PDP(m)-*Nal* for 20 min, always at 37°C.

## Transfection, lysis, and (co-)immunoprecipitation

For transient transfection cells were plated 1 d before to reach 40–50% confluency on the day of transfection. In case of multiple time points, cells were plated according to the expected confluency to reach 90% at the time of harvest and transfected in respective to that. All plasmid transfections were carried out using FuGene HD transfection reagent (Promega) and followed the manufacturer's instructions using OptiMEM (Gibco) and a ratio of plasmid: transfection reagent (wt/vol) of 1:6. For immunoprecipitation (IP) experiments followed by Western blotting or MS analysis, 2.5 $\mu$g of DNA were used for transfection in a cell culture plate with a diameter of 10 cm and expressed for 24 h. For siRNA transfection of *PPP5C* (s134722) pre-designed Silencer Select RNA (Thermo Fisher Scientific) was used and for PPP1C variants pre-designed siRNA

*PPP1CA/PPP1CB/PPP1CC* (Eurofins) obtained. All siRNA transfections were carried out using Lipofectamine RNAiMAX transfection reagent and followed the manufacturer's instructions. For lysis, cells were washed twice on ice with cold PBS and after addition of 500 µl lysis buffer (100 mM NaCl, 10 mM Tris, 0.1% IGEPAL, 1 mM EDTA, 1 mM EGTA, 20 nM Calyculin A, 1x cOmplete Mini protease inhibitor cocktail (Sigma-Aldrich), 1 mM DTT, 1 mM MnCl$_2$) cells were scraped from plates, lysed by 10 pushes through an injection needle (21 G, BD) and the insoluble fraction was pelleted by centrifugation (10 min, 10,000 rcf, 4°C). Supernatants were then used for in lysate dephosphorylation screens. Samples used for co-immunoprecipitation were incubated with anti-FLAG M2 magnetic beads (Merck Millipore) for 2 h at 4°C at the rotation wheel. Then washed three times with lysis buffer and boiled in 1x SDS sample buffer for 5 min at 95°C.

## Phosphatase depletion and protein degradation inhibition

Cell culture plates were pre-coated with fibronectin at a concentration of 1.25 µg/cm$^2$ (F2006; Sigma-Aldrich) before seeding with A7r5 cells. Cells were incubated for 24 h in cell culture incubator to allow for attachment. The next day, cells were transfected with siRNA targeting PP5 (s134722; Thermo Fisher Scientific) to deplete the protein. Medium was replaced 18 h post-transfection, and cells were then incubated in fresh medium for an additional 30 h. Subsequently, cells were treated with 100 nM Bafilomycin A1 (BafA1; Selleckchem) for 7 h at 37°C. Control plates received DMSO treatment under the same conditions. Cell harvesting was performed followed by mechanical lysis using a previously described lysis buffer and protein amounts were monitored via immunoblotting (20 µg cell lysate/lane) as described in the respective section.

## Phosphatase treatment

For phosphatase treatment of cell lysates, cells were lysed as described above, and endogenous Ser/Thr-phosphatases were inhibited by the addition of 20 nM CalA. Protein concentration was then determined by a BCA assay or via NanoDrop, and equal amounts of 1 mg total protein were incubated with recombinant PP1c/PP5 or buffer at a final phosphatase concentration of 1 µM in a total volume of 200 µl for up to 1 h (as indicated) at 30°C. Lysates were then immediately placed on ice. The reaction was terminated by the addition of 2x sample buffer and sample boiled for 5 min at 95°C. For on-bead dephosphorylation, we subjected the samples to immunoprecipitation using anti-FLAG M2 magnetic beads (Merck Millipore) for 2 h at 4°C. Beads were washed twice with lysis buffer and twice with reaction buffer (100 mM NaCl, 10 mM Tris, 1 mM MnCl$_2$, pH 7.5) prior incubation with 1 µM PP1c/PP5 for 30 min at 30°C in reaction buffer at 200 rpm on a thermoshaker (Thermomixer comfort; Eppendorf).

## On-bead digestion

For BAG3-targeted phosphoproteomics (TPP), 3FLAG-tagged BAG3 samples were washed twice with 50 mM ammonium bicarbonate (AmBiC) solution and incubated with sequencing grade trypsin (Promega) in a 1:50 trypsin-protein ratio (w/w) for 16 h at 37°C and

200 rpm on a thermoshaker (Thermomixer comfort; Eppendorf). Supernatants were acidified to a final concentration of 1% trifluoro acetic acid (TFA), dried in vacuo, and stored at –80°C until used for LC-MS anaylsis.

Pull-down samples from the BAG3 phospho-mutant variants (wt, 4A, and 4D) were transferred into buffer containing 2 M urea/50 mM AmBiC. Subsequently, proteins were reduced using 10 mM Tris(2-carboxy-ethyl)phosphine (TCEP) for 20 min at RT, followed by alkylation with 50 mM iodoacetamide (30 min at 37°C and 800 rpm on a thermoshaker in the dark) and tryptic digestion as described above at 800 rpm on a thermoshaker. Eluted peptides were cleaned up by C18-based desalting. For high pH reversed-phase stage-tip fractionation, peptides were resuspended in 50 µl 10 mM ammonia and loaded onto a µSpin cartridge (Affinisept). Peptides were eluted in eight sequential elution steps with 10 mM ammonia buffer containing 0%, 2.7%, 5.4%, 9%, 11.7%, 14.4%, 22.5%, or 64.8% ACN, respectively. Fractions 1 and 6, 2 and 7, 3 and 8 were combined and acidified with 20% TFA to reach a pH of 4. The resulting five samples were dried, desalted and reconstituted in 0.1% (vol/vol) TFA again before LC-MS measurement.

## Protein gel electrophoresis and immunoblotting with quantification

Samples were loaded onto NuPAGE 4–12% Bis-Tris protein gels (Invitrogen) and gel electrophoresis was carried out at a constant voltage of 180 V in 1x MOPS buffer. After electrophoresis, samples were transferred onto Immobilon-FL PVDF blotting membranes (Merck Millipore) using a wet blotting procedure. The membranes were blocked with Tris-buffered saline with 0.1% Tween (TBS-T) containing 5% (wt/vol) non-fat milk at RT for 60 min. After three washes (5 min), the membranes were incubated overnight at 4°C with the following antibodies in TBS-T with 5% BSA: $\alpha$-GAPDH (1:1,000, #2118), $\alpha$-BAG3 (1:1,000, 23842S), and $\alpha$-PP5 (1:1,000, 2289S), $\alpha$-CDC37 (1:1,000, 3618S), $\alpha$-CDC37-pS13 (1:1,000, 13,248), $\alpha$-phosphoThr (1:500, 42H4, 9386), SQSTM1/p62 (1:1,000, 5114T), and $\alpha$-14-3-3$\gamma$ (1:500, 5522S) obtained from Cell Signaling Technologies. Isoform-specific PP1 antibodies $\alpha$-PP1$\alpha$/$\beta$/$\gamma$ (1:1,000, sc-271762/sc-365678/sc-515943) were from Santa Cruz. The generic $\alpha$-phospho-Serine (1:500, AB1603) was obtained from Merck. The $\alpha$-HspB8 (1:1,000, STJ24102) antibody was purchased from the St. Johns Laboratory. $\alpha$-BAG3-pS136 was custom made by Eurogentec and its specificity was validated in (Ottensmeyer et al, 2024). After the overnight incubation, the blots were washed for 2 h in TBS-T and then incubated with donkey anti-rabbit IgG-horseradish peroxidase (HRP) conjugate (1:2,500, GENA934; GE Healthcare), goat anti-Guinea Pig IgG (H+L) (1:2,500, # A18769; Thermo Fisher Scientific), or rabbit anti-sheep IgG (H + L) HRP conjugate (1:2,500, 61-8620; Thermo Fisher Scientific) for 1 h at RT. After final three washes with TBS-T, the blots were developed using Western Lightning Plus-ECL Enhanced Chemiluminescence Substrate (Perkin Elmer). The blots were imaged using a Fusion FX Imaging System (Vilber) using the auto-analysis tool. Images were then analyzed by ImageJ (1.53K), background signals were subtracted and ratios of the phospho-protein signals to the respective total protein signals were calculated. Ratios were again normalized to the control sample and statistical tests were performed with GraphPad Prism v6.0.

Correlation was calculated using the implemented function of nonparametric Spearman correlation in GraphPad Prism. Contrast was enhanced in the illustrative blots to better visualize effects, following the journal's guidelines.

### In-gel digestion

After co-IP, protein sample was separated through protein gel electrophoresis. The gel was then stained using colloidal Coomassie Brilliant Blue and cut into 16 slices per sample lane. Next proteins were reduced in gel using TCEP at a concentration of 5 mM dissolved in 10 mM AmBiC for 30 min at 56°C, followed by the alkylation of free thiol groups with 50 mM 2-chloroacetamide (CAM) per 10 mM AmBiC at 37°C in the dark for 30 min. The proteins were then in-gel digested using trypsin (0.05 $\mu$g per slice) at 37°C overnight, followed by the desalting of peptides using C18 Stage-Tips. The peptides were lyophilized and reconstituted in 0.1% TFA.

### LC-MS analysis (TPP, in-gel digestion)

Reversed-phase liquid chromatography-mass spectrometry was performed using the UltiMateTM 3000 RSLCnano system (Dionex LC Packings/Thermo Fisher Scientific) coupled online to a Q Exactive Plus (Thermo Fisher Scientific) instrument. The UHPLC system was equipped with two C18 pre-columns ($\mu$PAC trapping column, PharmaFluidics) and a C18-endcapped analytical column (50 cm $\mu$PAC column, PharmaFluidics). Peptide separation and elution were generally performed at 40°C using a binary solvent system composed of 0.1% (vol/vol) FA/4% (vol/vol) (solvent A) and 86% (vol/vol) ACN/0.1% (vol/vol) FA (solvent B). Peptide mixtures from in-gel digestion were analyzed by LC-MS using a 1 h LC gradient. When loaded at 1% B with a flow rate of 10 $\mu$l*min$^{-1}$, peptides were eluted in a two stepped gradient starting after 5 min at 5% B and a flow rate of 0.3 $\mu$l*min$^{-1}$ over 24% B at min 25 reaching 42% B after 36 min. Subsequently, the column was flushed with 95% B for 5 min. The The MS instruments were externally calibrated using standard compounds and equipped with a nanoelectrospray ion source (Thermo Fisher Scientific). Parameters for mass spectrometric measurements on the Q Exactive in data-dependent acquisition (DDA) were as follows: MS scans, 375–1,700 m/z; resolution, 70,000 (at m/z 200); automatic gain control, 3 × 10$^6$; max. IT, 60 ms. Multiply charged peptide ions were fragmented by higher-energy collisional dissociation applying a normalized collision energy of 28% and a dynamic exclusion time of 45 s. A TOP12 method was applied to analyze peptides at a MS2 resolution of 35,000, an automatic gain control of 7 × 10$^2$, and a max. IT of 120 ms.

### LC-MS analysis (BAG3 phospho-mutants)

Peptides were analyzed by reversed-phase LC-MS using an UltiMate 3000 RSLCnano system (Thermo Fisher Scientific) coupled online to a Q Exactive Plus instrument (Thermo Fisher Scientific). The LC system was equipped with C18 pre-columns (nanoEase trapping column; Waters) and a C18 analytical column (nanoEase M/Z HSS C18 T3 75 × 250 mm column; Waters). Peptide separation and elution were performed at 40°C using a binary solvent system composed of 0.1% (vol/vol) formic acid (FA) (solvent A) and 80% (vol/vol) ACN in 0.1% (vol/vol) FA (solvent B). Peptides were analyzed by LC-MS using a 2-h method. Peptides were loaded at 1% solvent B for 3 min at a flow rate of 10 $\mu$l*min-1 and eluted by applying the following gradient: 1% 5% B in 2 min, 5% 37% B in 105 min, 37% 45% B in 2 min, followed by 11 min at 90% B and re-equilibration of the column for 11 min at 99% A. The flow rate for peptide elution was set to 0.3 $\mu$l*min-1. The MS instrument was equipped with a nano electrospray ion source and a coated emitter (EM-20-360-5pk; MicroOmics Technologies LLC). Parameters for mass spectrometric measurements in data-dependent acquisition mode were as follows: MS full scan window of m/z 375–1,200; resolution of 70,000 (at m/z 200); AGC of 3 × 106; maximum IT of 128 ms. Multiply charged peptide ions were fragmented by higher-energy collisional dissociation applying a normalized collision energy of 28% and a dynamic exclusion time of 45 s. A TOP15 method was applied to analyze peptides with an MS2 resolution of 17,500, an AGC of 1 × 105, and a maximum IT of 64 ms. Precursor isolation on the quadrupole was carried out with a 2.4 m/z window.

### Targeted MS

Dried peptide mixtures from on-bead digestion were resolved in 0.1% (vol/vol) TFA and iRT standard peptide mixture (Biognosys) in an 1:20 ratio. They were each analyzed applying a 2-h LC gradient and stepped higher-energy collisional dissociation (HCD) with a NCE of 20, 25, 30% (Diedrich et al, 2013). The gradient started at 5% B after 5 min with a flow rate of 0.3 $\mu$l*min$^{-1}$, 22% B after 70 min, and reached 40% after 105 min, before fushing for 3 min at 90% B. To setup an inclusion list for a targeted MS experiments by parallel reaction monitoring (PRM) samples were first run in DDA mode and analyzed using the software Skyline daily (22.2 64 bit) (MacLean et al, 2010; Pino et al, 2020). The inclusion list contained m/z values and retention time information of 14 Bag3 peptides and 9 phosphopeptides as well as the 11 iRT standard peptides (Biognosys). Peptide charge states 2–4 were included as identified by the DDA run and aloud for multiplexing in MS2. Whereas MS1 acquisition parameters were not changed from DDA, MS2 resolution was set to 17,500 (at m/z 200), an automatic gain control of 2 × 10$^5$, and a max. IT of 50 ms. Peptides from the inclusion list were selected for MS2 fragmentation in a 2 m/z isolation window and an MSX count of 3.

### Database searches and peptide spectrum analysis

Rawfiles were analyzed using MaxQuant software (version 2.2.0.0 for BAG3/PP5 pull-down experiments, version 2.0.2.0 for targeted MS experiments, version 2.4.4.0 for BAG3 phospho-mutant pull-down experiments) equipped with the Andromeda search engine (Cox & Mann, 2008; Cox et al, 2011). For database search, an in silico tryptic digest of a human proteome from the UniProt sequence database was used (pull-down experiments: downloaded Feb. 2023). For the targeted phosphoproteomics experiment just FLAG-tagged BAG3 and iRT fusion protein was used as database.

MaxQuant data processing was performed using the following parameters: variable modifications, S/T/Y phosphorylation (just S/T phosphorylation for the TPP experiment, no phosphorylation for BAG3 phospho-mutant pull-down experiments), M oxidation, and N-term acetylation; fixed modification, C carbamidomethylation;

first search mass tolerance, 20 ppm; false discovery rate, 1%; precursor ion mass tolerance, 4.5 ppm; fragment ion mass tolerance, 20 ppm; enzyme, trypsin/P; peptide charge states, +1 to +7; and "match between runs," enabled. Reverse, potential contaminant, and "only identified by site" hits were removed. To analyze data of the targeted MS experiments, the inclusion list, the MaxQuant results file msms.txt, PRM raw files, and the fasta file from the MQ search were imported into Skyline (MacLean et al, 2010). Peptide settings were as follows: protease, trypsin/P; max. missed cleavages, two; time window, 5 min; peptide length, 8–25 amino acids; modifications, cysteine carbamidomethylation, methionine oxidation, and S/T phosphorylation; max. variable modifications, three; and max. neutral loss, one. Default orbitrap transition settings were applied. Peptides with low quality, interfered MS1, and/or number of MS2 <4 were discarded manually. Extracted ion chromatograms for all peptides were manually inspected for correct peak picking, and peak integrations were adjusted if needed. For the identification of the Bag3 phospho-isoforms MQ peptides were filtered for an PTM-score > 0.95 and retention time information matched with fragment ion traces in skyline. Total MS1 extracted ion chromatogram areas were exported, and intensities were normalized to the sum intensity of the iRT peptides. Ratios for each phospho-isoform to the non-modified peptide were calculated for each replicate and condition to compare differences.

### Bioinformatic analysis

For the enrichment analysis of the BAG3 or PP5 IP, MaxQuant total protein group intensities of each replicate were then shifted to the summed mean intensities of each group. The intensities were subsequently normalized by variance stabilizing data transformation (Huber et al, 2002) and the rank sum method (Breitling & Herzyk, 2005), as implemented in the R package "RankProd" (v 3.11) (Del Carratore et al, 2017), was applied to calculate *P*-values for the enrichment of proteins that were quantified in all replicates. We generated a protein–protein interaction network (PPIN) for the 235 significant genes using the StringApp (Doncheva et al, 2019) integrated into Cytoscape (Shannon et al, 2003), with interactions having a confidence score of 0.4 or higher. To evaluate the network properties, we used the NetworkAnalyzer (Assenov et al, 2008) plugin within Cytoscape. This plugin computes various network metrics for each node, including degree and betweenness centrality, among others. In addition, to further explore the modular organization of the PPIN, we applied the Molecular Complex Detection (MCODE) (Bader & Hogue, 2003) plugin in Cytoscape. This clustering analysis identified densely interconnected regions within the PPIN, helping to unveil potential functional modules and intricate relationships between genes. To elucidate the functional relevance of the clusters, we conducted String enrichment analysis. This analysis aimed to identify the gene ontology terms associated with the nodes, providing insights into the potential functions and processes in which these genes are involved.

For the analysis of PP5 binding to the BAG3 variants, the MaxQuant proteinGroup table was first cleaned from proteins only identified by site, contaminants, and reverse hits. Each replicate was then normalized by shifting the summed MS intensity of all protein groups per replicate was shifted to the mean of the summed MS intensities across all samples. Subsequently, PP5/BAG3 intensity ratios of BAG3 and PP5 were calculated. A *t* test was performed to determine *P*-values between different BAG3 mutants.

### BAG3 conservation analysis

For the multiple sequence alignment of BAG3 orthologs, we harvested the NCBI database for all available sequences (NCBI, Feb'23) and filtered for sequences being not predicted and not of low quality. Only one isoform for each species was used. The alignment was performed with the Geneious software, using the implemented MUSCLE algorithm (Edgar, 2004) for multiple protein alignments. For graphical representation of the alignment (including identity/ conservation and sequence logos), all alignment segments corresponding to gaps in the reference sequence (*homo sapiens* BAG3) were removed. Identity scores were calculated by the MUSCLE algorithm. The consensus tree was built with the implemented Tree builder plugin in Geneious using the Jukes-Contor model and Neighbor-Joining method after 200 bootstrap replications. The consensus tree for the 377 orthologs was grouped into the three most comprehensive taxa groups (excluding unassigned orthologs and amphibians), when disregarding the same hierarchical level of taxonomic classification for the groups.

### Gene ontology and overrepresentation analysis

Additional gene ontology information for enriched protein sets of the BAG3 and PP5 co-IPs was obtained from UniProt Consortium et al (2021). Quantification of GO biological process terms was conducted with the R software (R Core Team, 2020) with additional packages tidyverse (v2.0.0) (Wickham et al, 2019), ggplot2 (v3.4.2) (Wickham, 2016), and UniprotR (v2.2.0) (Soudy et al, 2020). Statistical overrepresentation test of co-enriched genes was carried out with the implemented gene list analysis tool from the PANTHER database (v17.0) (Thomas et al, 2022). PANTHER GO-Slim biological process annotations were compared with overall representation of *homo sapiens* after the protocol of the publisher (Mi et al, 2019). Statistical significance was determined using Fisher's test (*P*-value < 0.05) with the Bonferroni correction.

# Data Availability

Mass spectrometry proteomics data have been deposited in PRIDE repository with the dataset identifier: PXD050006; PXD050035; and PXD057392.

# Supplementary Information

# Acknowledgements

This work was supported by the German Research Foundation DFG (FOR2743 project ID 388932620 to J Höhfeld, M Köhn, and B Warscheid) (BIOSS EXC 294 to M Köhn), and funded by a Consolidator grant of the European Research Council (ERC) to M Köhn (#865119).

## Author Contributions

T Kokot: formal analysis, validation, investigation, visualization, methodology, and writing—original draft, review, and editing.
JP Zimmermann: formal analysis, investigation, visualization, methodology, and writing—review and editing.
Y Chand: formal analysis, visualization, and methodology.
F Krier: investigation and methodology.
L Reimann: investigation and methodology.
L Scheinost: investigation and methodology.
N Höfflin: investigation and methodology.
A Esch: investigation.
J Höhfeld: conceptualization, resources, funding acquisition, and writing—review and editing.
B Warscheid: resources, supervision, funding acquisition, and writing—review and editing.
M Köhn: conceptualization, resources, supervision, funding acquisition, methodology, project administration, and writing—review and editing.

## Conflict of Interest Statement

The authors declare that they have no conflict of interest.

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
