## [Reviewer comments · Life Science Alliance]

Life Science Alliance

Identification of phosphatases that dephosphorylate the co-chaperone BAG3

Thomas Kokot, Johannes Zimmermann, Yamini Chand, Fabrice Krier, Lena Reimann, Laura Scheinost, Nico Höfflin, Alessandra Esch, Joerg Hoehfeld, Bettina Warscheid, and Maja Köhn

DOI: <https://doi.org/10.26508/lsa.202402734>

Corresponding author(s): Maja Köhn, University of Bonn

Review Timeline:

Submission Date:	2024-03-25
Editorial Decision:	2024-06-07
Revision Received:	2024-10-31
Editorial Decision:	2024-11-01
Revision Received:	2024-11-06
Accepted:	2024-11-07

Transaction Report:

June 7, 2024

Re: Life Science Alliance manuscript #LSA-2024-02734-T

Prof. Maja Köhn
University of Freiburg
Biology
Schänzlestrasse 18
Freiburg 79104
Germany

Dear Dr. Köhn,

Thank you for submitting your manuscript entitled "Identification of phosphatases that dephosphorylate the co-chaperone BAG3" to Life Science Alliance. The manuscript was assessed by expert reviewers, whose comments are appended to this letter. We invite you to submit a revised manuscript addressing the Reviewer comments.

Thank you for this interesting contribution to Life Science Alliance. We are looking forward to receiving your revised manuscript.

Sincerely,

B. MANUSCRIPT ORGANIZATION AND FORMATTING:

Reviewer #1 (Comments to the Authors (Required)):

In this manuscript the role of protein phosphatases working on BAG3 is investigated. BAG3 is involved in protein elimination pathways and highly important for several human diseases including Alzheimers. Therefore it is important to understand how BAG3 is regulated. Although previous work have revealed that BAG3 phosphorylation is of importance this has not been investigated carefully and it is unclear what the phosphatases acting on BAG3 are. Here the authors combine mass spectrometry with cellular and biochemical assays to investigate how PP1 and PP5 protein phosphatases regulate BAG3 phosphorylation status. They provide evidence that PP1 dephosphorylates S136 while PP5 dephosphorylates a cluster of phosphorylation sites (S284/T285/S289/S291). Collectively the experiments are nicely controlled and the manuscript easy and interesting to read. I have a few suggestions for improvements that I think would make the manuscript more strong.

- 1) The phosphoproteomic analysis they conduct on BAG3 with PP5 modulation could be interesting to conduct with PP1 activity modulation to determine if PP1 regulates additional sights in BAG3.
- 2) It would be important to validate the BAG3 phosphoantibody they use by showing it does not recognize the mutant form of BAG3 (S136A). I could not find this in the manuscript.
- 3) It would be interesting to know if the PP5-BAG3 interaction is regulated by phosphorylation of the S284/T285/S289/S291 cluster. Could the authors do a co-IP of BAG3 WT and mutant and look at PP5 binding.

Reviewer #2 (Comments to the Authors (Required)):

In this manuscript, the authors did a pull-down of BAG3 and identified two potential phosphatases that are responsible for the dephosphorylation of a specific phosphosite (S134) and a cluster around T285. the work is cleanly done and the manuscript is well written. The manuscript will be of interest to everyone working in the fields where BAG3 plays an important biological role. the main weaknesses I identified are that 1. the in vitro dephosphorylation is not a great proof as subunits, that often provide the substrate specificity, are not present. 2. it is not shown for PP5 if the enzyme is actually active - an active control/substrate would be great to show that PP5 is indeed not phosphatase for these p-sites.

Reviewer #3 (Comments to the Authors (Required)):

"Identification of Phosphatases that Dephosphorylate the Co-Chaperone BAG3"

The authors sought to identify phosphatases which act on functional phosphorylation sites for the BAG3 co-chaperone. Using multiple approaches including mass spectrometry analysis of proteins co-immunoprecipitating with Flag-tagged BAG3 or PP5, biochemical, cellular biochemical, pharmacological, and knock-down of endogenous phosphatases, they identify that PP1 and PP5 function to dephosphorylate BAG3.

The article is well-written and clear. The experimental approaches and data analysis were strong. The conclusions are quite well supported by their data.

Major point:

The functional effect of dephosphorylating BAG3 by PP1 and PP5 is not fully characterized to a level that would merit publication in Life Science Alliance. The authors could address this by establishing a cell-based assay that was responsive to the phosphorylation state of BAG3. In this assay, the authors could use inhibitors, knockdowns, and BAG3 phosphorylation site mutants (S136A; T285A; T289A; and possibly others in the cluster), to assess the functional consequences of PP1- and PP5-dependent dephosphorylation of BAG3.

However, that said, two molecular effects of BAG3 dephosphorylation are identified/characterized to some degree and in some settings:

Reduced 14-3-3 gamma binding to BAG3 with CalA application

Increased BAG3 protein levels with siRNA application to knockdown endogenous phosphatases

These could serve as one possible starting point for further functional consequences of BAG3 dephosphorylation, particularly in conjunction with BAG3 mutants that cannot be phosphorylated at relevant residues. However, the authors could focus on another cellular function driven by BAG3 phosphorylation, as elucidated in other studies. Given the novel result of B above, however, some characterization, or at least a proposed model, of BAG3 levels increasing based on knocking down phosphatases is needed.

Additional points:

1. In the abstract: I believe "BAG3-p136" should "BAG3-pS136"...
2. How many MS biological and/or technical replicates were performed for the co-IPs involving BAG3 and PP5? I may have missed this, but please indicate in the legends the replicate numbers whatever they might be.
3. For Fig. 2: do the authors have evidence that OA was functioning? For example, did they probe cell extracts with a more generic anti-pS/T motif antibody and observe loss of signal relative to loading controls?
4. For Fig. 5: Representative MS/MS spectra showing distinguishing features of each phosphopeptide and its dephosphorylation should be included in the supplementary material.
5. In the Discussion: Second paragraph, "As starting point..." should be "As a starting point..."
6. In the Discussion: A more expanded discussion of the kinases for these sites and their co-regulation of the sites with PP1 and PP5 integrated into a model would improve the discussion. Related to this, if any relevant kinases or kinase localization proteins were found in the proteomics analysis they should be discussed. Also related, an additional column in the supplementary tables indicating kinase or phosphatase functionality (including subunits or kinase/phosphatase docking proteins) would be particularly helpful.

Response to reviewers' comments:

Authors: We would like to thank the editor and all reviewers for their thorough and critical evaluation of our work and the constructive suggestions.

Reviewer #1 (Comments to the Authors (Required)):

In this manuscript the role of protein phosphatases working on BAG3 is investigated. BAG3 is involved in protein elimination pathways and highly important for several human diseases including Alzheimers. Therefore it is important to understand how BAG3 is regulated. Although previous work have revealed that BAG3 phosphorylation is of importance this has not been investigated carefully and it is unclear what the phosphatases acting on BAG3 are. Here the authors combine mass spectrometry with cellular and biochemical assays to investigate how PP1 and PP5 protein phosphatases regulate BAG3 phosphorylation status. They provide evidence that PP1 dephosphorylates S136 while PP5 dephosphorylates a cluster of phosphorylation sites (S284/T285/S289/S291). Collectively the experiments are nicely controlled and the manuscript easy and interesting to read. I have a few suggestions for improvements that I think would make the manuscript more strong.

1) The phosphoproteomic analysis they conduct on BAG3 with PP5 modulation could be interesting to conduct with PP1 activity modulation to determine if PP1 regulates additional sights in BAG3.

Authors: Phosphoproteomic analyses similar to the one requested have previously been conducted with PP1. Incubation of PP1c with HeLa cell lysates resulted in the dephosphorylation of BAG3 at serine residues 173, 275, and 289 (human BAG3, pS136 was not detected at all in this setting). The observed log2-fold changes of phosphorylation relative to control were -0.287, -1.065, and -1.459, respectively (Hoermann et al. 2020). However, activation of PP1c in cells through the activating peptide PDP-Nal (5 mins, HeLa cells) resulted in no significant dephosphorylation of BAG3 p-sites in our recent study (Hoermann et al. 2024). Only pS377 showed slightly elevated phosphorylation levels, indicating secondary effects on this site. We therefore concluded in the Hoermann et al. 2024 study that the previously in vitro detected sites (Hoermann et al. 2020) were not high confidence substrate sites in cells.

2) It would be important to validate the BAG3 phosphoantibody they use by showing it does not recognize the mutant form of BAG3 (S136A). I could not find this in the manuscript.

Authors: The antibody was validated in the publication Ottensmeyer et al. 2024, which has recently been published. We now refer to this publication in the Materials and Methods section in our manuscript.

3) It would be interesting to know if the PP5-BAG3 interaction is regulated by phosphorylation of the S284/T285/S289/S291 cluster. Could the authors do a co-IP of BAG3 WT and mutant and look at PP5 binding.

Authors: We thank the reviewer for this interesting idea. We have carried out the following experiment:

Mutation of pCMV-3FLAG-BAG3 into an 4x Ala and 4x Asp mutant form

- 1.) pCMV-3FLAG-BAG3- S284A/T285A/S289A/S291A (BAG3_4A)
- 2.) pCMV-3FLAG-BAG3- S284D/T285D/S289D/S291D (BAG3_4D)

Upon overexpression and co-IP followed by LC-MS/MS to analyze PP5 binding, we found that PP5 bound stronger to the 4D variant than to the wild-type protein, but weaker to the 4A variant, suggesting that PP5 binding is regulated by the negative charges that are displayed on phosphorylated BAG3 as the substrate. The results are shown in Figure 5 D and presented in the respective results chapter:

“After confirming that PP5 dephosphorylates the BAG3 p-site cluster both in vitro and in cells, we inquired whether phosphorylation of BAG3 in this cluster would regulate PP5 binding to BAG3. To this end, we co-precipitated overexpressed FLAG-tagged wild-type (wt), S284A/T285A/S289A/S291A (4A) and S284D/T285D/S289D/S291D (4D) BAG3 variants and measured PP5 binding by MS read out (Fig 5D). We observed that BAG3_4D bound significantly stronger to PP5 than BAG3_4A, with the wt variant - being in part phosphorylated (Fig 5C) - binds to PP5 with an intermediate efficacy, suggesting that the negative charges support PP5 binding.”

Figure 5D

Reviewer #2 (Comments to the Authors (Required)):

In this manuscript, the authors did a pull-down of BAG3 and identified two potential phosphatases that are responsible for the dephosphorylation of a specific phosphosite (S134) and a cluster around T285.

the work is cleanly done and the manuscript is well written. The manuscript will be of interest to everyone working in the fields where BAG3 plays an important biological role. The main weaknesses I identified are that

1. the in vitro dephosphorylation is not a great proof as subunits, that often provide the substrate specificity, are not present.

Authors: We agree with Reviewer #2 that relying solely on in vitro dephosphorylation studies may not provide conclusive evidence due to the absence of specific subunits that grant substrate specificity. Recognizing this limitation, we included cellular dephosphorylation experiments, for PP1 in particular with a selective inhibitor and a selective activator, which offer a more comprehensive view of the biological processes involved.

Regarding BAG3-pS136, we followed up here on the two most prominent regulatory subunits identified in BAG3 co-immunoprecipitation assays: MYPT1 and FKBP15. Depletion of these proteins did not result in increased phosphorylation (see below Western blots), suggesting that their involvement in dephosphorylating BAG3-pS136 may be limited or absent, or masked through cellular adaption by the long time periods.

Western blots: *Two regulatory subunits, which were found in the FLAG-BAG3 co-IP, were depleted. However, the knockdown did not result in an increase of the pS136 levels compared to total BAG3.*

Quantification of WBs of FKBP15 or MYPT1 knockdown (n=4).

One hypothesis is that PP1c might directly interact with BAG3, potentially identifying BAG3 as a new regulatory subunit for PP1c. If this were the case, the binding site needs to be identified. However, this is very challenging to address due to the absence of a typical binding motif (such as the RVxF motif) known to bind to PP1, which would give at least hints toward the binding site. This absence does not exclude BAG3 from being an interacting protein of PP1, as there are other proteins that bind to PP1 without a typical binding motif (for example SDS22). As mentioned above, the time scale difference between the knockdown (1-2 days) and signalling events (seconds to minutes) could also lead to a rescue by other regulatory subunits. Therefore, identifying the responsible regulatory subunit, be it BAG3 itself or another protein, is outside the scope of this work.

For the BAG3 phosphorylation site cluster (pS284, pT285, pS289, pS291), we conducted experiments in cells by overexpressing protein phosphatase 5 (PP5), which led to dephosphorylation of BAG3. Importantly, PP5 does not act as holoenzyme with regulatory subunits, but it is known to bind via the TPR domain to other proteins. In this regard, a recent study by Devi et al. (2024) demonstrates that PP5 can interact with a variety of binding partners, suggesting mechanisms for chaperone (HSP70/90)-independent activation of PP5, which supports our observations (Devi et al. 2024).

We discuss these points now in the discussion section of the manuscript:

“In addition to the possibility of interacting with BAG3 through regulatory subunits, BAG3 may interact with PP1 directly in a transient manner not easily detectable by IPs. If so, identifying the binding site is challenging due to the absence of a typical binding motif, such as the RVxF motif, which could provide clues (Wakula et al. 2003; Bollen et al. 2010). However, this does not rule out BAG3 as a PP1 interactor, as some proteins, like SDS22, bind PP1 without such motifs (Bollen et al. 2010; Heroes et al. 2019). Therefore, the complex question of how exactly PP1 interacts with its substrate BAG3 will be part of future studies.”

And

“While PP5 dephosphorylated pS136 only to a very limited extent, it emerged as a strong phosphatase candidate for BAG3 due to their consistent co-occurrence in co-IPs (Fig. 1B, Fig. 4A). Interestingly, we did not observe the co-occurrence of HSP70/HSP90, well-known activators and substrate-directing

proteins of PP5, in both co-IPs. However, a recent study demonstrates that several other proteins can activate PP5 without the involvement of HSPs (Devi et al 2024).”

2. it is not shown for PP5 if the enzyme is actually active - an active control/substrate would be great to show that PP5 is indeed not phosphatase for these p-sites.

Authors: We detected very weak dephosphorylation kinetics of PP5 for BAG3-pS136 during the *in vitro* dephosphorylation of endogenous BAG3 over the course of 60 minutes (Fig. 2E), suggesting that it is an unsuitable substrate. Under similar conditions at the 60-minute time point using the MS read-out, we observed complete dephosphorylation of the BAG3 p-site cluster including amino acids 284-291 (Fig. 5C). Furthermore, overexpression of PP5 in cells for 24 hours did not result in the dephosphorylation of BAG3-pS136, but in the dephosphorylation of the p-site cluster (Fig. 5A and C). These results suggested to us that PP5 is active.

In addition, to validate the activity of recombinant PP5 in the requested setting we now tested its activity on its reported substrate site CDC37-pS13 as a positive control in the titration experiment over time, observing significant dephosphorylation already within 15 minutes (now Fig S2D) and confirming that PP5 is active. These kinetics are similar to those observed for PP1c towards BAG3-pS136 (Fig. 2E).

Supplementary Figure 2D. Dephosphorylation of CDC37-pS13 in lysate through incubation with PP5.

In the manuscript:

“Interestingly, PP5, the most enriched phosphatase in the BAG3 co-immunoprecipitation (co-IP) dataset, exhibited slower dephosphorylation kinetics for BAG3-pS136 compared to its known substrate p-site, pS13, on the Hsp90 co-chaperone Cdc37 (CDC37; Fig S2D) (Dushukyan et al. 2017).”

Reviewer #3 (Comments to the Authors (Required)):

"Identification of Phosphatases that Dephosphorylate the Co-Chaperone BAG3"

The authors sought to identify phosphatases which act on functional phosphorylation sites for the BAG3 co-chaperone. Using multiple approaches including mass spectrometry analysis of proteins co-immunoprecipitating with Flag-tagged BAG3 or PP5, biochemical, cellular biochemical, pharmacological, and knock-down of endogenous phosphatases, they identify that PP1 and PP5 function to dephosphorylate BAG3.

The article is well-written and clear. The experimental approaches and data analysis were strong. The conclusions are quite well supported by their data.

Major point:

The functional effect of dephosphorylating BAG3 by PP1 and PP5 is not fully characterized to a level that would merit publication in Life Science Alliance. The authors could address this by establishing a cell-based assay that was responsive to the phosphorylation state of BAG3. In this assay, the authors could use inhibitors, knockdowns, and BAG3 phosphorylation site mutants (S136A; T285A; T289A; and possibly others in the cluster), to assess the functional consequences of PP1- and PP5-dependent dephosphorylation of BAG3.

However, that said, two molecular effects of BAG3 dephosphorylation are identified/characterized to some degree and in some settings:

Reduced 14-3-3 gamma binding to BAG3 with CalA application

Increased BAG3 protein levels with siRNA application to knockdown endogenous phosphatases
These could serve as one possible starting point for further functional consequences of BAG3 dephosphorylation, particularly in conjunction with BAG3 mutants that cannot be phosphorylated at relevant residues. However, the authors could focus on another cellular function driven by BAG3 phosphorylation, as elucidated in other studies. Given the novel result of B above, however, some characterization, or at least a proposed model, of BAG3 levels increasing based on knocking down phosphatases is needed.

Authors: We acknowledge the major concern raised about the need for a deeper characterization of the functional effects of BAG3 dephosphorylation by PP1 and PP5 to meet the publication standards of Life Science Alliance. Overall, we have now deepened our findings by providing evidence that the dephosphorylation of BAG3 by PP5 facilitates the binding of HspB8 to BAG3 (new Figure 5E). Additionally, we examined the effects over a longer timescale and found that depletion of PP5 has similar effects on BAG3 as CASA inhibition. Moreover, PP5 depletion increases BAG3 sensitivity to CASA inhibition. This indicates that BAG3 cannot undergo CASA-mediated degradation when the p-site cluster is not dephosphorylated by PP5, thereby hindering the interaction with HspB8. This interaction is crucial for the functional BAG3 core complex in the CASA pathway.

In addition, in a very recent publication by our co-author Jörg Höfelf, the requested phosphorylation mutants were investigated as part of a larger study (Ottensmeyer et al. 2024). They functionally show that reduced phosphorylation of S136, T285, and T285/S289 activates the CASA degradation pathway, but they do not explore any phosphatase involvement. Here, we identify the corresponding phosphatases involved in this process, and report the major direct functional effects in that the binding

of 14-3-3 and of HspB8 to BAG3 are regulated by PP1 through pS136 and by PP5 through the p-site cluster at amino acids 284-291, respectively, affecting BAG3 protein levels in the context of CASA.

We have added Figure 6 for an overall view on the findings and proposed functions. Our findings provide a significant and detailed connection between the identification of selected BAG3 p-sites, their regulation, and their documented roles in processes such as protein transport (Xu et al. 2013), cell division (Luthold et al. 2021), and response to mechanical force (Ottensmeyer et al. 2024). We now discuss particularly the recent publication (Ottensmeyer et al. 2024) in more detail in the manuscript in the Discussion section.

Additional points:

1. In the abstract: I believe "BAG3-p136" should "BAG3-pS136"...

Authors: Thank you for catching this mistake. We corrected it.

2. How many MS biological and/or technical replicates were performed for the co-IPs involving BAG3 and PP5? I may have missed this, but please indicate in the legends the replicate numbers whatever they might be.

Authors: We accordingly indicated the replicate numbers (n=4) in the figure captions.

3. For Fig. 2: do the authors have evidence that OA was functioning? For example, did they probe cell extracts with a more generic anti-pS/T motif antibody and observe loss of signal relative to loading controls?

Authors: We appreciate Reviewer #3's concern. Using the remaining samples from three available replicates (see source data), we performed immunoblots with generic anti-pS/pT antibodies. We have added one blot to the main figure (Fig. 2A) to demonstrate the functionality of our experiment and to show that BAG3-pS136 is responsive to CalA but not to OA.

Adjusted Figure 2A with the generic pSer/pThr blot.

In the manuscript we wrote:

"The overall efficacy of the phosphatase inhibitors was verified using a generic pS/pT antibody (Fig 2A)."

4. For Fig. 5: Representative MS/MS spectra showing distinguishing features of each phosphopeptide and its dephosphorylation should be included in the supplementary material.

Thank you for this suggestion. In response, we have added one representative MS/MS spectrum for each phosphopeptide to the source data file of Figure 5, as Supplementary Figure 5 is already quite detailed. The newly added spectra are displayed below.

5. In the Discussion: Second paragraph, "As starting point..." should be "As a starting point..."

Authors: Thank you for catching this mistake. We corrected it.

6. In the Discussion: A more expanded discussion of the kinases for these sites and their co-regulation of the sites with PP1 and PP5 integrated into a model would improve the discussion. Related to this, if any relevant kinases or kinase localization proteins were found in the proteomics analysis they should be discussed. Also related, an additional column in the supplementary tables indicating kinase or phosphatase functionality (including subunits or kinase/phosphatase docking proteins) would be particularly helpful.

Authors: Thank you for your valuable feedback. In response, we have expanded the supplementary table for the BAG3 co-IP to include additional details, specifically highlighting identified phosphatases, kinases, and their regulatory proteins. We nevertheless want to indicate that the focus of this work is

on phosphatases. These candidates and their regulators are highlighted in Figure 2D as well as in the updated Supplementary Table 1 of the Bag3 co-IP.

In total, we identified 11 significant hits in the co-IP dataset that suggest kinase activity (indicated now in the Supplementary Table 1). However, none correspond to previously reported kinases acting on BAG3. Nonetheless, our dataset serves as a valuable source for future studies exploring BAG3 regulation through kinases, which we now also mention in the last paragraph of the discussion. It is important to note that our PPI findings are based on an uninduced/unstimulated setup without external stimuli, such as cellular stress or cell cycle arrest, which may account for the absence of reported kinase activity at the two previously identified sites.

As mentioned in the manuscript, the kinase responsible for phosphorylation at BAG3-pS136 has yet to be identified. Although previous studies have identified CDK1 (T285; Luthold et al. 2021) and PKC δ (S187; Li et al. 2013) as *in vivo* kinases for BAG3 at different sites, these kinases were not detected in our proteomics data of the BAG3 co-IP. Our findings, however, demonstrate that PP5 dephosphorylates pT285, positioning it as a functional antagonist to CDK1 at this site. Additionally, other regulatory proteins previously reported, such as CDK5R1 (S291) and SQSTM1 (T285) (Luthold et al. 2021), were not detected in our dataset. A recent study showed that Sequestosome 1 (SQSTM1/p62) is needed for T285 phosphorylation at least during mitosis (Luthold et al. 2021). Moreover, p62 participates in CASA through its interaction with BAG3 (Gamerdinger et al. 2009; Guilbert et al. 2018). SQSTM1 is part of the CASA machinery and plays a role in BAG3 degradation. Thus, we now investigated SQSTM1 protein levels upon PP5 depletion combined with CASA inhibition (Fig S5F). Depletion of PP5, which limits PP5's ability to dephosphorylate BAG3, also results in SQSTM1 accumulation similar to BAG3 accumulation (Fig 5E, Fig S5F). While a trend for SQSTM1 is observed with BafA1 treatment, it is not significant without simultaneous PP5 depletion (Fig S5F). Upon depletion of PP5, SQSTM1 showed increased sensitivity to BafA1 treatment. These findings suggest that PP5 depletion might hinder CASA progression due to the lack of the ability to dephosphorylate BAG3, and lead to the accumulation of CASA components. However, to verify this it requires further investigation, which is beyond the scope of this study. We now describe these new results (Figures 5G and S5F) and discuss them in the manuscript at the respective sections in the results and discussion chapters.

References

- Devi, S., Charvat, A., Millbern, Z., Vinueza, N. and Gestwicki, J. E.** (2024). Exploration of the binding determinants of protein phosphatase 5 (PP5) reveals a chaperone-independent activation mechanism. *The Journal of biological chemistry* **300**, 107435.
- Gamerding, M., Hajieva, P., Kaya, A. M., Wolfrum, U., Hartl, F. U. and Behl, C.** (2009). Protein quality control during aging involves recruitment of the macroautophagy pathway by BAG3. *The EMBO journal* **28**, 889–901.
- Guilbert, S. M., Lambert, H., Rodrigue, M.-A., Fuchs, M., Landry, J. and Lavoie, J. N.** (2018). HSPB8 and BAG3 cooperate to promote spatial sequestration of ubiquitinated proteins and coordinate the cellular adaptive response to proteasome insufficiency. *FASEB journal : official publication of the Federation of American Societies for Experimental Biology* **32**, 3518–3535.
- Hoermann, B., Dürr, E.-M., Ludwig, C., Ercan, M. and Köhn, M.** (2024). A strategy to disentangle direct and indirect effects on (de)phosphorylation by chemical modulators of the phosphatase PP1 in complex cellular contexts. *Chemical science* **15**, 2792–2804.
- Hoermann, B., Kokot, T., Helm, D., Heinzlmeir, S., Chojnacki, J. E., Schubert, T., Ludwig, C., Berteotti, A., Kurzawa, N., Kuster, B. et al.** (2020). Dissecting the sequence determinants for dephosphorylation by the catalytic subunits of phosphatases PP1 and PP2A. *Nature communications* **11**, 3583.
- Li, N., Du, Z.-X., Zong, Z.-H., Liu, B.-Q., Li, C., Zhang, Q. and Wang, H.-Q.** (2013). PKC δ -mediated phosphorylation of BAG3 at Ser187 site induces epithelial-mesenchymal transition and enhances invasiveness in thyroid cancer FRO cells. *Oncogene* **32**, 4539–4548.
- Luthold, C., Lambert, H., Guilbert, S. M., Rodrigue, M.-A., Fuchs, M., Varlet, A.-A., Fradet-Turcotte, A. and Lavoie, J. N.** (2021). CDK1-Mediated Phosphorylation of BAG3 Promotes Mitotic Cell Shape Remodeling and the Molecular Assembly of Mitotic p62 Bodies. *Cells* **10**.
- Ottensmeyer, J., Esch, A., Baeta, H., Sieger, S., Gupta, Y., Rathmann, M. F., Jeschke, A., Jacko, D., Schaaf, K., Schiffer, T. et al.** (2024). Force-induced dephosphorylation activates the cochaperone BAG3 to coordinate protein homeostasis and membrane traffic. *Current biology* **34**, 4170-4183.
- Xu, Z., Graham, K., Foote, M., Liang, F., Rizkallah, R., Hurt, M., Wang, Y., Wu, Y. and Zhou, Y.** (2013). 14-3-3 protein targets misfolded chaperone-associated proteins to aggresomes. *Journal of cell science* **126**, 4173–4186.

November 1, 2024

RE: Life Science Alliance Manuscript #LSA-2024-02734-TR

Prof. Maja Köhn
University of Freiburg
Biology
Schaenzlestrasse 18
Freiburg 79104
Germany

Dear Dr. Köhn,

Thank you for submitting your revised manuscript entitled "Identification of phosphatases that dephosphorylate the co-chaperone BAG3". We would be happy to publish your paper in Life Science Alliance pending final revisions necessary to meet our formatting guidelines.

- please be sure that the authorship listing and order is correct
- please add a callout for Figure S5D to your main manuscript text
- the datasets deposited into PRIDE should be made publicly accessible at this point, removing the need for Reviewer access information in the Data Availability statement

A. FINAL FILES:

B. MANUSCRIPT ORGANIZATION AND FORMATTING:

**Submission of a paper that does not conform to Life Science Alliance guidelines will delay the acceptance of your

manuscript.**

The license to publish form must be signed before your manuscript can be sent to production. A link to the electronic license to publish form will be available to the corresponding author only. Please take a moment to check your funder requirements.

Sincerely,

November 7, 2024

RE: Life Science Alliance Manuscript #LSA-2024-02734-TRR

Prof. Maja Köhn
University of Bonn
Institute for Cell Biology
Ulrich-Haberland Str. 61a
Bonn 53121
Germany

Dear Dr. Köhn,

Thank you for submitting your Research Article entitled "Identification of phosphatases that dephosphorylate the co-chaperone BAG3". It is a pleasure to let you know that your manuscript is now accepted for publication in Life Science Alliance. Congratulations on this interesting work.

DISTRIBUTION OF MATERIALS:

Again, congratulations on a very nice paper. I hope you found the review process to be constructive and are pleased with how the manuscript was handled editorially. We look forward to future exciting submissions from your lab.

Sincerely,
